# BCL-X$_L$ directly modulates RAS signalling to favour cancer cell stemness

Sophie de Carné Trécesson [1,8], Frédérique Souazé[1], Agnès Basseville[1], Anne-Charlotte Bernard[1], Jessie Pécot[1], Jonathan Lopez [2], Margaux Bessou[2], Kristopher A. Sarosiek[3], Anthony Letai[3], Sophie Barillé-Nion[1], Isabelle Valo[4,5], Olivier Coqueret[5,6], Catherine Guette[5,6], Mario Campone[7], Fabien Gautier[1,7] & Philippe Paul Juin[1,7]

In tumours, accumulation of chemoresistant cells that express high levels of anti-apoptotic proteins such as BCL-X$_L$ is thought to result from the counter selection of sensitive, low expresser clones during progression and/or initial treatment. We herein show that BCL-X$_L$ expression is selectively advantageous to cancer cell populations even in the absence of pro-apoptotic pressure. In transformed human mammary epithelial cells BCL-X$_L$ favours full activation of signalling downstream of constitutively active RAS with which it interacts in a BH4-dependent manner. Comparative proteomic analysis and functional assays indicate that this is critical for RAS-induced expression of stemness regulators and maintenance of a cancer initiating cell (CIC) phenotype. Resistant cancer cells thus arise from a positive selection driven by BCL-X$_L$ modulation of RAS-induced self-renewal, and during which apoptotic resistance is not necessarily the directly selected trait.

[1] Team 8 "Stress adaptation and tumor escape", CRCINA - Institut de Recherche en Santé de l'Université de Nantes-Angers - IRT, BP 70721, 8 quai Moncousu, Nantes 44007, France. [2] Service de Biochimie et Biologie moléculaire—Centre Hospitalier Lyon Sud, Faculté de Médecine Lyon Sud—Université Lyon 1, Centre de Recherche en Cancérologie de Lyon—INSERM U1052 CNRS U5286, Lyon 69003, France. [3] Dana-Farber Cancer Institute, Harvard Medical School, Boston, MA 02215, USA. [4] Biopathology Department, ICO - Centre de Lutte contre le Cancer Paul Papin, 15 rue André Boquel, Angers, France. [5] Team 12 'Targeted Therapies and Tumor Escape in Colorectal Cancer', CRCINA - Institut de Recherche en Santé de l'Université de Nantes-Angers - Centre de Lutte contre le Cancer Paul Papin, 15 rue André Boquel, Angers 49055, France. [6] ICO site Paul Papin, 15 rue André Boquel, Angers 49055, France. [7] ICO site René Gauducheau, Boulevard Jacques Monod, Saint Herblain 44805, France. [8] Present address: Oncogene Biology Laboratory, The Francis Crick Institute, 1 Midland Road, London NW1 1AT, UK. Correspondence and requests for materials should be addressed to P.P.J. (email: philippe.juin@univ-nantes.fr)

Anti-apoptotic proteins of the BCL-2 family (BCL-2, BCL-X$_L$ or MCL-1) are frequently up-regulated in cancers as a result of genetic, epigenetic or signalling pathway changes[1]. BCL-2 homologues negatively regulate mitochondrial outer membrane permeabilisation (MOMP) and promote cell survival by counteracting death signals that result from direct activation of their pro-apoptotic multi-domain counterparts (BAX/BAK) by «activator» BH3-only proteins (BIM, BID or PUMA). They do so by sequestering the BH3 domains of pro-apoptotic proteins. BCL-2, BCL-X$_L$ and MCL-1 display complementary survival activities as they interact with overlapping but distinct, differentially regulated, pro-apoptotic partners. BCL-X$_L$ has the more potent anti-apoptotic activity as it binds to the widest spectrum of pro-apoptotic counterparts. Its over-expression correlates with chemoresistance in cancer cell lines[2] and in triple negative breast cancer patient samples[3]. This underscores the potential interest of BH3-mimetics inhibitors of BCL-X$_L$ in chemo-resistant cancers. Pro-apoptotic inhibitors of BCL-X$_L$ nevertheless have a narrow therapeutic window and thus the eradication of high BCL-X$_L$ expresser cancer cells remains difficult to achieve with such compounds[4].

Mechanisms that drive the outgrowth of high BCL-X$_L$ expressing cells are not fully characterized. The current consensus is that BCL-X$_L$ provides a survival advantage to cancer cells under apoptotic pressures induced discontinuously by therapy or continuously by oncogenic alterations. MOMP is indeed the primary way by which cancer cells die in response to radiotherapy, chemotherapy and to diverse stress stimuli cancer cells encounter as tumours progress[5]. MOMP is also part of an intrinsic tumour suppressor mechanism induced by oncogenic alterations that lead to aberrant expression of C-MYC or loss of the pRB tumour suppressor[6]. These types of aberrations impose a sustained cell-autonomous pressure that should select cancer cells with higher levels of BCL-X$_L$. However, not all oncogenic signals increase the apoptotic load of cancer cells and in some cases oncogene activity alleviates it instead. RAS activity for instance inhibits apoptosis[7].

RAS pathway activation frequently occurs in solid tumours as a result of direct RAS mutations or of other less direct causes e.g. downstream of EGFR stimulation/activation[8]. Importantly, RAS activity features have been described in the absence of RAS mutations in triple negative breast cancers[9]. Activation of RAS and its downstream pathways MAPK/ERK and PI3K/AKT have well documented anti-apoptotic consequences due to the induction of anti-apoptotic proteins expression and the down-regulation or inactivation of pro-apoptotic effectors[7]. The latter effect should decrease the pressure to select for cancer cells with enhanced expression of BCL-2 homologs. This raises the question of what, if any, selective advantage BCL-X$_L$ overexpression brings to cancer cells in RAS-activated tumours, and more generally in tumours that are not in receipt of an apoptotic pressure.

In addition to survival maintenance, other biological effects have been reported for BCL-X$_L$ and ascribed to its ability to interact with proteins beyond the BCL-2 family. BCL-X$_L$ may thus positively regulate biological functions contributing to tumour growth and dissemination by modulating the activity of some components of its vast interactome[10]. How critical and advantageous such regulations would be to RAS-driven cancer cells and the binding partners involved in this context remain largely unknown. Dysregulated RAS activation induces a plethora of signalling pathways that favour cell proliferation, motility and invasion. In mammary epithelial cells, it promotes an epithelial to mesenchymal transition (EMT) and the emergence of cancer initiating cells (CICs) endowed with self-renewal capacities[11, 12]. CICs regenerate new tumours after an initial regression and play a critical role in tumour progression, in particular after treatment, to which they resist better than non-CICs[13, 14]. The influence of RAS activity on phenotypic plasticity and on the dynamic equilibrium between non-CICs and CICs therefore plays a key role in the expansion of epithelial tumour cell populations, initially or after relapse. We show that BCL-X$_L$ contributes to this process by interacting directly with RAS and fine-tuning its downstream activity.

## Results

**BCL-X$_L$ is required for RAS-induced CIC phenotype.** To explore the biological functions of BCL-X$_L$ in transformed epithelial cells we used mammary epithelial MCF10A cells stably transduced with KRAS$^{V12}$ cDNA retroviral vectors[15]. These cells are endowed with enhanced phenotypic plasticity and CIC properties. Indeed, MCF10A KRAS$^{V12}$ cells express mesenchymal markers (Supplementary Fig. 1a, e) and a subset of these cells form mammospheres and express enhanced levels of the typical CD44 marker (Supplementary Fig. 1b, c. Please also see Supplementary Methods and Supplementary Note 1 for more details). MCF10A KRAS$^{V12}$ cells exhibited a decrease in *BCL2L11* mRNA (BIM) expression and enhanced *BCL2L1* (BCL-X$_L$) and *MCL1* mRNA expressions compared to controls (Fig. 1a). *BBC3* (PUMA), *PMAIP1* (NOXA), *BAX* or *BAK* mRNAs were in contrast expressed to similar levels in our matched pair of cell lines and *BCL2*, *BCL2L2* and *BCL2A1* expressions were barely detectable. *BCL2L11* and *BCL2L1* expressions in MCF10A KRAS$^{V12}$ cells are regulated by ongoing RAS activity as judged by the effects of RAF inhibition on their expression (Fig. 1b). Western blot analysis showed no detectable change in MCL-1 proteins levels, perhaps owing to the innate lability of this protein. In contrast, BIM protein levels were down-regulated and BCL-X$_L$ protein levels were up-regulated in RAS-activated cells (Fig. 1c). Intracellular immunostaining of BCL-X$_L$ in MCF10A KRAS$^{V12}$ cells showed that these cells express BCL-X$_L$ levels according to a lognormal distribution and lack an obvious subpopulation of high BCL-X$_L$ expressing cells (Fig. 1d top). However, double immunostaining of BCL-X$_L$ and CD44 revealed that cells expressing the highest levels of BCL-X$_L$ encompass subpopulations with the highest expression of CD44 (Fig. 1d bottom). We suspected from this repartition that BCL-X$_L$ might impact on the CIC phenotype and we investigated this further.

Downregulation of BCL-X$_L$ by a lentivirus based sh-RNA approach (sh-BCL-X$_L$) had no impact on the viability of the bulk MCF10A KRAS$^{V12}$ population (Supplementary Fig. 2a, b). We did not detect any effect of BCL-X$_L$ knockdown on the overall doubling time of the population either (Supplementary Fig. 2c). In contrast, sh-BCL-X$_L$ diminished the percentage of mammosphere-forming cells (MFC) as strongly as sh-RNA knockdown of IL-6, a cytokine which plays a role in CIC maintenance[16]. Sh-RNA mediated BAX knockdown was used as a control for a possible impact of the RNAi machinery on MFC and we found it had no effect (Fig. 2a). To confirm that the effects of the three sh-BCL-X$_L$ used are on-target effects, we treated with one given sh-BCL-X$_L$ KRAS$^{V12}$-transformed cells after their infection with a lentivirus encoding for a sh-RNA resistant variant of BCL-XL cDNA. The resulting cells, in contrast to control cells, did not decrease their amount of mammosphere forming cells (Supplementary Fig. 2d). In an additional independent approach *BCL2L1* (BCL-X$_L$ gene) was knocked out using CRISPR/Cas9 in KRAS$^{V12}$ cells. Knock out cells showed a decrease in MFC compared to control cells (Supplementary Fig. 2e). Importantly, we confirmed the involvement of BCL-X$_L$ in mammosphere formation in second-generation assays in KRAS$^{V12}$ cells as well as in mammosphere formation in the KRAS wild type human breast cancer cell line MDA-MB-468 (Supplementary Fig. 2f, g). A role for BCL-X$_L$ in self-renewal was also found in a non-transformed context, since

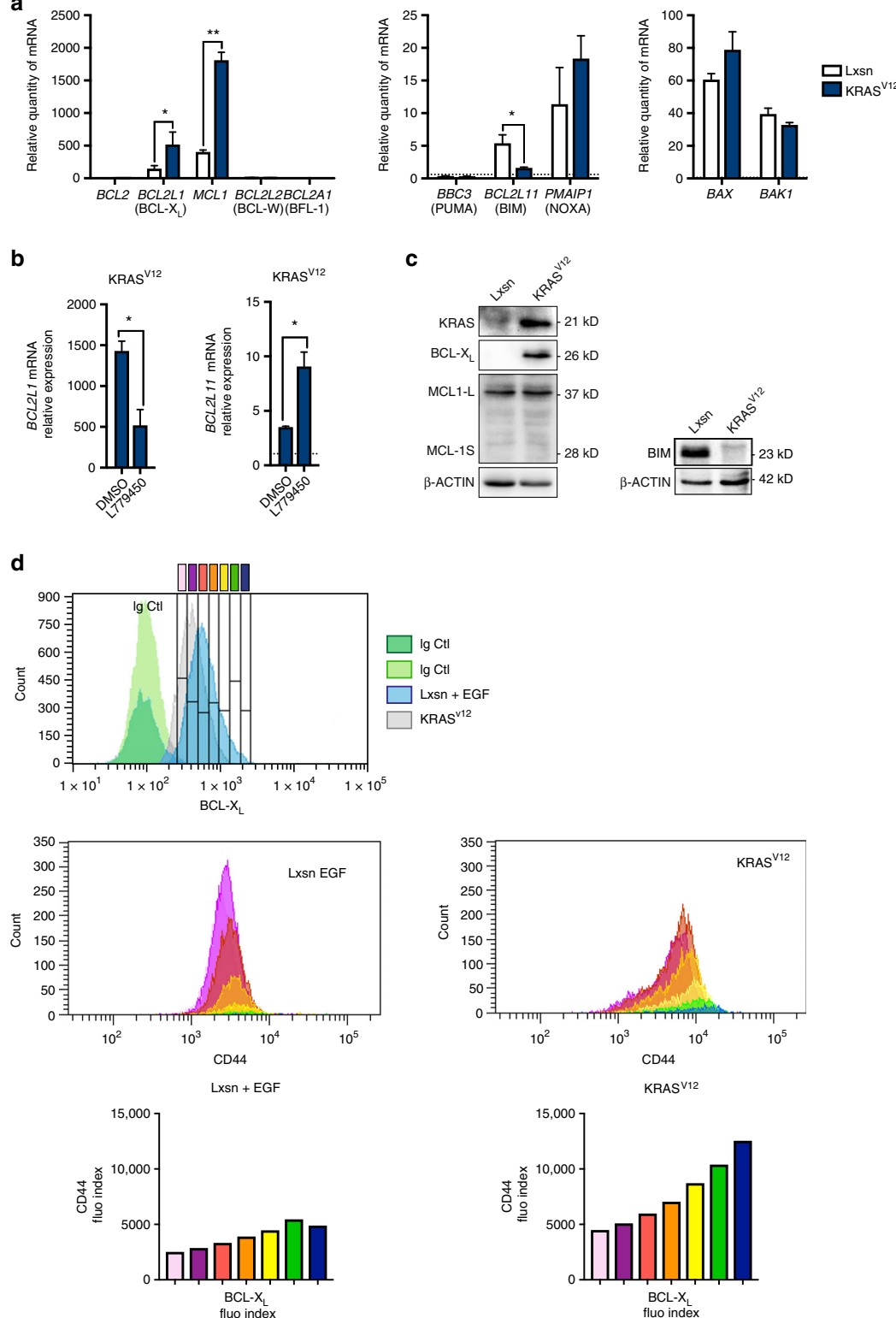

**Fig. 1** Oncogenic RAS induces BCL-XL expression correlated with enhanced CD44 expression. **a** qPCR analysis of BCL2 family member mRNA expression in MCF10A Lxsn and KRAS$^{V12}$ cell lines grown in adherent conditions. Mean and SEM of 3 independent experiments are represented as relative quantity of mRNA normalised to the mean of *RPLP0*, *RPS18* and *B2M* relative expression (two-tailed unpaired *t*-test). **b** Western blot showing KRAS, BCL-XL, MCL-1 and BIM expression in MCF10A Lxsn and KRAS$^{V12}$ cell lines. **c** qPCR analysis of *BCL2L1* and *BCL2L11* mRNA expressed in MCF10A KRAS$^{V12}$ cell line treated with 10 μM of RAF inhibitor (L779450). Mean and SEM of three independent experiments are represented as relative quantity of mRNA normalised to the mean of *RPLP0*, *RPS18* and *B2M* relative expression (two-tailed unpaired *t*-test). **d** Flow cytometry analysis of a co-staining for intracellular BCL-X$_L$ and cell surface CD44. Expression of CD44 (middle and bottom) is shown in populations of increasing levels of BCL-X$_L$ as evaluated in Top panel (grey, EGF treated Lxsn cells, blue, KRAS$^{V12}$ cell). Stainings with control isotypes are shown in green. Data from one representative experiment are shown

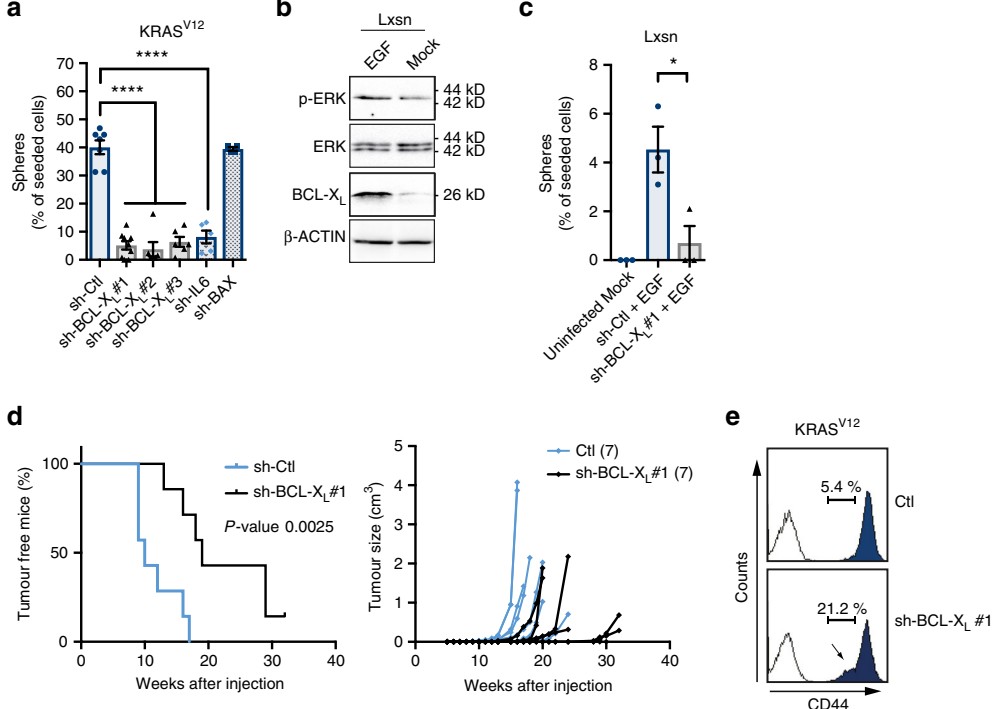

**Fig. 2** BCL-X$_L$ maintains cancer initiating cell features induced by RAS activity in mammary cells. **a**, **b** Percentage of sphere forming cells in bulk population of **a** MCF10A KRAS$^{V12}$ cells or **c** EGF-treated MCF10A Lxsn cells after 72 h infection with the indicated lentivector sh-RNAs. As comparison, the measured lack of sphere formation of Lxsn untreated with EGF is also illustrated. 128 cells per condition were seeded in serum-free media in ultra-low adhesion plates and the resulting spheres counted after two weeks incubation. Each dot represents the percentage of cell forming sphere for one biological replicate. Mean and SEM of at least 3 (3 to 6) independent experiments are represented (two-tailed unpaired t-test). **b** Western blot analysis of BCL-X$_L$ expression and ERK phosphorylation in MCF10A Lxsn cells after overnight starvation and treatment with 20 ng ml$^{-1}$ EGF for an additional 24 h **d**. Transplant experiment of MCF10A KRAS$^{V12}$ cells infected with sh-BCL-X$_L$ or a control vector. $5 \times 10^5$ cells were subcutaneously injected in 7 Nu/Nu mice for each group and tumour growth was monitored (Log-rank (Mantel-Cox) test). **e** Flow cytometry analysis of CD44 expression on MCF10A KRAS$^{V12}$ cells infected with sh-BCL-X$_L$ or a control vector 72 h prior staining. White histograms represent the IgG control staining; blue histograms represent the CD44 staining. Data from one representative experiment are shown

EGF treatment of MCF10A Lxsn induced BCL-X$_L$ expression (Fig. 2b) and increased the percentage of MFC in a BCL-X$_L$ dependent manner (Supplementary Fig. 1b and 2c). Moreover, overexpression of BCL-X$_L$ promoted EGF-induced sphere forming in MCF-7 cell line (see below). Altogether, these results highlight a role for BCL-X$_L$ in self-renewal in oncogenic and non-oncogenic RAS activated models.

We confirmed a role for BCL-X$_L$ in CIC induction by KRAS$^{V12}$ in vivo by evaluating the ability of a minimal number of BCL-X$_L$-depleted MCF10A KRAS$^{V12}$ cells to seed new tumours in immunodeficient mice (Fig. 2d left). BCL-X$_L$-deficient cells initiated tumours that seemed to grow with similar rates than control cells but with a significant delay (Fig. 2d right). This is consistent with an effect of BCL-X$_L$ on the initial number of tumour seeding cells and not on tumour progression per se.

We further examined if the canonic anti-apoptotic function of BCL-X$_L$ could explain its impact on CIC maintenance. No cell death rates were detected in KRAS$^{V12}$-induced mammospheres following sh-BCL-X$_L$ (Supplementary Fig. 3a). Caspase inhibition did not rescue mammosphere formation in these conditions and no effect on mammosphere formation was detected upon BH3 mimetic (ABT-737) treatment (Supplementary Fig. 3b, c). Altogether, these results indicate that the effect of BCL-X$_L$ on CIC representation does not ensue from an impact on CIC viability and does not rely on its canonical anti-apoptotic activity (See Supplementary Note 2 for more details). Instead, BCL-X$_L$ appears to directly regulate some features of CICs. Consistent with this, we found that sh-BCL-X$_L$ decreased the

representation of CD44$^{high}$ cells in the MCF10A KRAS$^{V12}$ population (Fig. 2e).

**BCL-X$_L$ supports RAS activation to induce HMGA2 and FOSL1 expression.** We performed iTraq labelling and quantitative mass spectrometric analysis of protein lysates from EGF-treated MCF10A Lxsn cells and MCF10A KRAS$^{V12}$ cells depleted or not in BCL-X$_L$[17]. We used four different labelling reagents to compare protein expressions between each of the different cell contexts (Supplementary Fig. 4. Please also see Supplementary Methods for more details). We thus identified proteins whose expression was induced by RAS activation or affected by BCL-X$_L$ in conditions promoting MFC (i.e. EGF-treated MCF10A and MCF10A KRAS$^{V12}$ cells) (Fig. 3a). We quantified expression of 2118 proteins in total. Proteins were defined as "BCL-X$_L$-dependent" when their fold change log2 expression was lower than -0.18 following BCL-X$_L$ knockdown in MCF10A KRAS$^{V12}$ cells or in EGF-treated MCF10A Lxsn cells (bottom 30% representing respectively 668 and 612 proteins). Proteins were called as "RAS-induced" when their fold change log2 expression was higher than 0.28 in MCF10A KRAS$^{V12}$ cells compared to EGF-treated Lxsn (top 30% representing 648 proteins). By this approach, we identified 118 proteins whose expression was "RAS-induced" and "BCL-X$_L$ dependent" in both cell lines (Table 1).

We identified CD44 expression as RAS-induced BCL-X$_L$-dependent (in line with the result shown in Fig. 2e) and KRAS

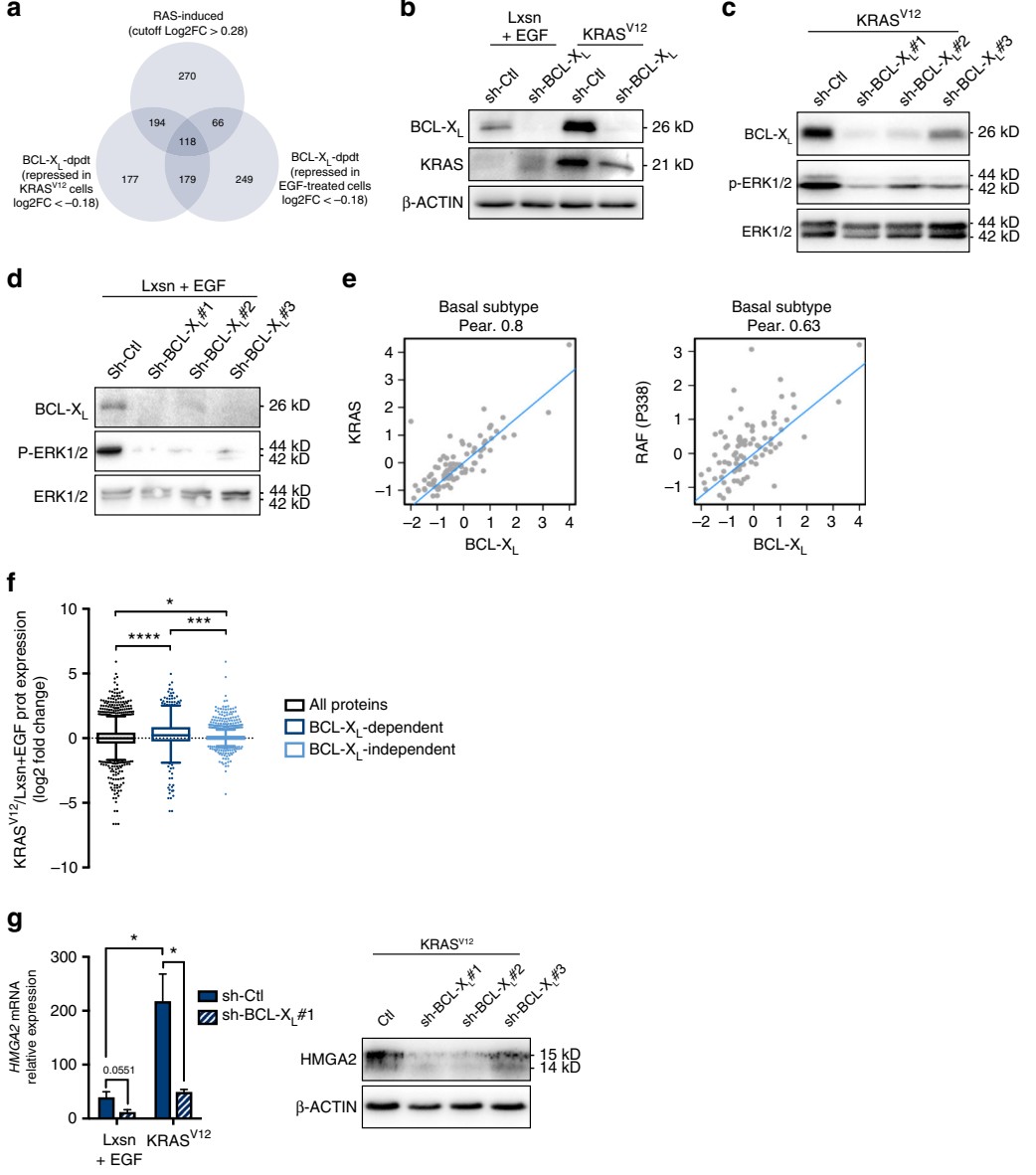

**Fig. 3** BCL-X$_L$ contributes to MAPK activation to induce HMGA2 expression. **a** Venn diagram of repartition of protein expression variations in iTraq experiment. Applied cut-offs on log2 Fold Change are mentioned for each condition. RAS-induced proteins are those for which the log2 Fold Change is higher than 0.28 in MCF10A KRAS$^{V12}$ sh-Ctl compared to MCF10A Lxsn sh-Ctl. BCL-X$_L$-dpdt proteins are those for which the Log2 fold change is lower than −0.18 in sh-BCL-X$_L$ compared to sh-Ctl in either EGF-treated MCF10A Lxsn cells (right) or MCF10A KRAS$^{V12}$ cells (left). **b–d** Western blot showing KRAS, BCL-X$_L$ and/or p-ERK expressions in **b** MCF10A Lxsn and KRAS$^{V12}$ cells, **c** MCF10A KRAS$^{V12}$ cells 72 h after sh-BCL-X$_L$ and **d** MCF10A Lxsn cells 72 h after sh-BCL-X$_L$ (in presence of 20 ng ml$^{-1}$ EGF in MCF10A Lxsn media). **e** Correlations between BCL-X$_L$ protein expression and KRAS protein expression (left) or p338RAF protein expression (right) in basal subtype tumour samples. Quantified expression of BCL-X$_L$, KRAS and p338RAF from RPPA data were examined for correlation using Pearson's (Pear.) analysis. The results shown here are based upon data generated by the TCGA Research Network: http://cancergenome.nih.gov/. **f** Box & Whiskers representation of KRAS$^{V12}$ vs. Lxsn log2 Fold Change protein expression (Tukey representation, unpaired $t$-test with equal SD). Black box represents log2FC for all proteins, dark blue box represents KRAS vs. Lxsn log2FC of BCL-X$_L$-dependent proteins (as defined above), light blue box represents KRAS vs. Lxsn log2FC of BCL-X$_L$-independent proteins (with sh-BCL-X$_L$ vs. sh-Ctl log2FC <−0.18 or >0.18 in KRAS$^{V12}$ background). **g** qPCR analysis of *HMGA2* mRNA in MCF10A Lxsn and KRAS$^{V12}$ cell lines infected with sh-BCL-X$_L$ during 72 h (in presence of 20 ng ml$^{-1}$ EGF in MCF10A Lxsn media). Mean and SEM of 3 independent experiments are represented as relative quantity of mRNA normalised to the mean of *RPLP0*, *RPS18* and *GAPDH* relative expression (two-tailed unpaired t-test). Western blot analysis showing *HMGA2* expression in MCF10A Lxsn cell line infected with sh-BCL-X$_L$ during 72 h

itself. Down-regulation of KRAS upon BCL-X$_L$ depletion was confirmed in MCF10A KRAS$^{V12}$ cells by western blot analysis (Fig. 3b) and was confirmed in MCF10A KRAS$^{V12}$ cells in which *BCL2L1* was knocked out by CRISPR/Cas9 (Supplementary Fig. 5a). We assume this reflects a positive effect of BCL-X$_L$ on KRAS protein turnover, since sh-BCL-X$_L$ had no detectable

impact on *KRAS* mRNA expression (Supplementary Fig. 5b), whereas it enhanced the ability of cycloheximide (a protein synthesis inhibitor) to decrease KRAS protein levels (Supplementary Fig. 5c). RAS proteins have been reported to be subject to proteasomal[18] and/or lysosomal[19] degradation. BCL-X$_L$ appears to protect RAS from the latter and not from the former,

**Table 1 BCL-X$_L$-dependent RAS-induced proteins**

| | | | |
|---|---|---|---|
| ABCC3 | DYNC1I2 | MAP1B | RPN2 |
| ACO1 | EIF2AK2 | ME1 | RPS27A |
| ACTR1A | EIF4G3 | MOGS | RRAS |
| ACTR3 | ENAH | MRI1 | SCARB2 |
| AHSA1 | ERGIC1 | MSN | SDCBP |
| AP2A2 | ESYT1 | MTCH2 | SERPINH1 |
| AP2S1 | ETFB | MVP | SFXN3 |
| ARL6IP5 | FKBP3 | MYOF | SH3BGRL |
| ARPC5 | FMNL2 | NEBL | SHC1 |
| ATP1B1 | GABARAPL2 | NIPSNAP3A | SLC16A3 |
| ATP6V1A | GALNT2 | NNMT | SLC38A2 |
| B4GALT1 | GFM1 | NPC1 | SLC9A3R2 |
| BCL2L13 | GFPT2 | NPEPPS | SNRPG |
| BSG | GLUD1 | NUCB1 | SOD2 |
| C1QBP | GPAA1 | OGDH | SQRDL |
| CALR | HMGA2 | P4HB | SSR4 |
| CAPG | HNRNPUL2 | PARP4 | STT3B |
| CAPRIN1 | HYOU1 | PC | SUGT1 |
| CD44 | IDH3B | PDCD6 | SULT1A3 |
| CFL2 | IDI1 | PDHA1 | SURF4 |
| CKAP4 | IKBKG | PDIA3 | THOC2 |
| CNPY2 | IMPDH1 | PLIN3 | TIMM44 |
| COPA | IQGAP3 | PLOD1 | TM9SF4 |
| CRIP2 | ITGA3 | PNPO | TMED1 |
| CS | ITGA5 | PON2 | TMEM173 |
| CSTB | KLC1 | PRDX1 | TXNRD1 |
| CYB5R3 | KRAS | PSMD8 | USP4 |
| DDOST | KYNU | PTRF | VASP |
| DECR1 | LAMB1 | PYCR1 | |
| DUSP23 | LMAN2 | RPN1 | |

List of proteins found in the MS analysis whose are common elements of "RAS-induced", "BCL-X$_L$-dpdt (KRAS$^{V12}$)" and "BCL-X$_L$-dpdt (Lxsn)" illustrated and described in Fig. 3a: RAS-induced proteins are those for which the log2 Fold Change is higher than 0.28 in MCF10A KRAS$^{V12}$ sh-Ctl compare to MCF10A Lxsn sh-Ctl. BCL-X$_L$-dpdt proteins are those for which the Log2 fold change is lower than −0.18 in sh-BCL-X$_L$ compare to sh-Ctl in either EGF-treated MCF10A Lxsn cells or MCF10A KRAS$^{V12}$ cells

as bafilomycin A1 and chloroquin, but not MG132, enhanced KRAS proteins levels in BCL-X$_L$ knock out cells (Supplementary Fig. 5d). Importantly, the effect of BCL-X$_L$ on KRAS expression levels coincides with an effect on downstream signalling, as BCL-X$_L$ depletion led to decreased phosphorylation of ERK (p-ERK) in MCF10A KRAS$^{V12}$ cells and in EGF-treated MCF10A cells (Fig. 3c, d).

We investigated whether the impact of BCL-XL on RAS expression and activity was globally found in human breast tumours. Analysis of reverse-phase protein array (RPPA) data of human basal-like breast cancer samples from The Cancer Genome Atlas revealed that BCL-X$_L$ expression was correlated with KRAS and with phosphorylation levels of c-RAF at S338 that mark RAS activation (Fig. 3e). The same correlations were observed in other breast cancer subtypes (Supplementary Fig. 5e, f). In contrast, BCL-2 expression was correlated neither with KRAS nor with phospho-c-RAF in any subtype (Supplementary Fig. 5g, h). By immunochemistry analysis, we could observe a tendency in BCL-X$_L$ expression to correlate with p-ERK in an independent cohort of 108 triple negative breast cancer samples (Supplementary Fig. 5i, j). Altogether, these clinical results are consistent with mutual influences between BCL-X$_L$ and KRAS activity in human breast cancers.

Our iTraq analysis indicated that BCL-X$_L$-induced proteins tended to be more frequently induced by RAS than the whole set or the BCL-X$_L$-independent set (Fig. 3f. Please also see Supplementary Methods for more details). This further underlines the contribution of BCL-X$_L$ to RAS downstream signals. Ingenuity pathway analysis revealed that our list of the 118 RAS-

induced BCL-X$_L$-dependent proteins was enriched for positive targets of TGFβ signalling (Supplementary Data 1), which can cooperate with RAS signalling to favour phenotypic plasticity and induce a CIC phenotype[20]. These included proteins described to regulate cell adhesion and invasion (such as ITGA5, ITGA3 and MSN) and the embryonic factor HMGA2. The identification of the latter as a RAS-induced BCL-X$_L$-dependent protein is particularly relevant due to its established role as a transcriptional regulator of EMT and self-renewal downstream of both TGFβ and RAS[18–20]. We confirmed that BCL-X$_L$ knockdown decreased HMGA2 expression not only at the protein but also at the mRNA levels (Fig. 3g).

The dependency on BCL-X$_L$ of some RAS targets might directly result from the fact that they require BCL-X$_L$-driven full RAS signalling to be expressed. This predicts that their expression would be sensitive to even a mild decline in RAS signal intensity. Consistent with this, *HMGA2* mRNA expression in MCF10A KRAS$^{V12}$ cells was highly sensitive to inhibition by low concentrations of the RAF kinase inhibitor L779450 (Fig. 4a). In contrast, expressions of *BCL2L1* (BCL-X$_L$), *BCL2L11* (BIM) and *CCND1* (a cell cycle regulator induced by the numerous transcription factors that RAS activates) were only affected at higher concentrations. Similar results were obtained using a concentration range of the MEK inhibitor U0126 (Supplementary Fig. 6a) as well as in EGF-treated MCF10A Lxsn cells (Fig. 4b and Supplementary Fig. 6b). We reasoned that, reciprocally, the expression of genes that are highly sensitive to inhibition of RAS signalling should show BCL-X$_L$ dependency. Expression of *FOSL1*, which encodes for a transcription factor that contributes to breast CIC maintenance, is highly sensitive to changes in MAPK/ERK signalling[21, 22]. Consistently, its expression was higher in MCF10A KRAS$^{V12}$ cells than in EGF-treated MCF10A Lxsn cells and inhibited in both models by low doses of L779450 (Fig. 4a–c) and U0126 (Supplementary Fig. 6b). In both models Sh-BCL-X$_L$ decreased *FOSL1* expression while it had no significant effect on *CCND1* expression (Fig. 4c). This argues that BCL-X$_L$ is necessary for RAS-induced expression of self-renewal regulators (HMGA2, FOSL1) because it supports a fully active signalling pathway downstream of stabilized RAS.

**BCL-X$_L$ interacts with KRAS to favour downstream signalling**. BCL-2 homologues exert their biological functions by modulating the activity of numerous binding partners. We sought for protein interactants of BCL-X$_L$ in MCF10A KRAS$^{V12}$ cells by immunoprecipitation of BCL-X$_L$ followed by mass spectrometric analysis and identified KRAS as a putative binding partner for BCL-X$_L$ (Supplementary Data 2). Interactions between KRAS and BCL-X$_L$ were confirmed by co-immunoprecipitation assays from MCF10A KRAS$^{V12}$ lysates and by pull down assays with recombinant BCL-X$_L$ and KRAS (Fig. 5a). To further attest that BCL-X$_L$/KRAS interactions occur in a whole live cell context, and to ensure that they were not artificially favored by detergents used in the above assays, we performed Bioluminescence Resonance Energy Transfer (BRET) experiments (See Supplementary Note 3 for more details). Saturable BRET signals were observed between increasing levels of YFP-fused to the N-terminal end of wild type BCL-X$_L$ and R-Luc fused to the N-terminal end of KRAS. This was observed regardless of KRAS mutational status and thus GTP-binding state (Fig. 5b). BRET signals between BCL-X$_L$ and KRAS were neither inhibited by a single mutation in BCL-X$_L$ that affects its BH3 binding (the G138A substitution, Fig. 5c) nor by treatment with BH3 mimetics ABT-737 or WEHI-539 which inhibited BRET signals between BCL-XL and the BH3-only protein tBID (Supplementary Fig. 7). This indicates that the BCL-X$_L$ BH3 binding interface, on which relies its canonic anti-

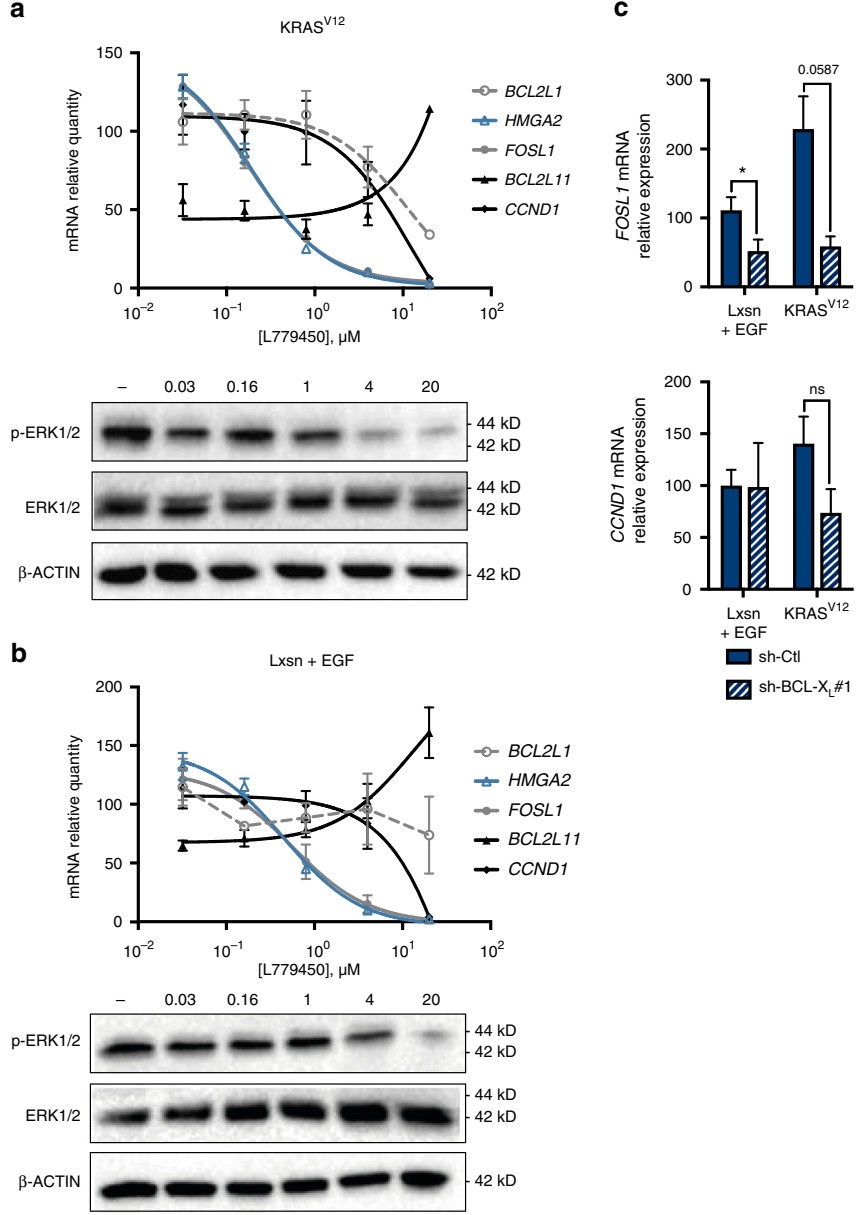

**Fig. 4** BCL-X$_L$-dependent RAS target genes expression are sensitive to low dose of RAF inhibitor. **a** qPCR analysis of *BCL2L1*, *HMGA2*, *FOSL1*, *BCL2L11* and *CCND1* mRNA in MCF10A KRAS$^{V12}$ cells treated with increasing doses of RAF inhibitor (L779450) during 24 h. Mean and SEM of 3 independent experiments are represented as relative quantity of mRNA normalised to the mean of *RPLP0*, *RPS18* and *ACTB* relative expression. Insert: western blot showing phosphorylation of ERK and total ERK levels under the same conditions. **b** qPCR of *BCL2L1*, *HMGA2*, *FOSL1*, *BCL2L11* and *CCND1* mRNA in MCF10A Lxsn cells grown in the presence of EGF treated with increasing doses of RAF inhibitor (L779450) during 24 h. Mean and SEM of 3 independent experiments are represented as relative quantity of mRNA normalised to the mean of *RPLP0*, *RPS18* and *GAPDH* relative expression Insert: western blot showing phosphorylation of ERK and total ERK levels under the same conditions. **c** qPCR of *FOSL1* and *CCND1* mRNA in EGF-treated MCF10A Lxsn and in MCF10A KRAS$^{V12}$ cell lines infected with sh-BCL-X$_L$ during 72 h. Mean and SEM of 3 independent experiments are represented as relative quantity of mRNA normalised to the mean of *ACTB*, *HPRT1* and *GAPDH* relative expression (two-tailed unpaired t-test)

apoptotic function, is not directly involved. In contrast, deletion in the BH4 domain that plays a critical role in BCL-X$_L$ interactions with numerous partners outside of the BCL-2 family[10], significantly diminished BRET signals (Fig. 5c).

To confirm that BH4-dependent interactions between KRAS and BCL-X$_L$ account for its effect on RAS signalling, we used a BRET-based RAS activity sensor in epithelial human breast cancer MCF-7 cells stably overexpressing equivalent levels of wild type or BH4-deleted BCL-X$_L$. This allowed us to monitor the influence of BCL-X$_L$ and its BH4 domain on RAS activation

kinetics following EGF addition. Overexpression of wild type BCL-X$_L$ enhanced the early response to EGF and significantly prolonged it but the BH4-deleted RAS binding deficient mutant failed to do so (Fig. 5d). In further support to a critical role for the BH4 domain of BCL-X$_L$ we observed that the overexpression of wild type BCL-X$_L$ but not the BH4-deleted mutant promoted EGF-induction of FOSL1 and HMGA2 in MCF-7 cells (Fig. 5e). Likewise, BH4-deleted BCL-X$_L$ was significantly less efficient than wild type BCL-X$_L$ to favour mammosphere formation by EGF-treated MCF-7 cells (Fig. 5f).

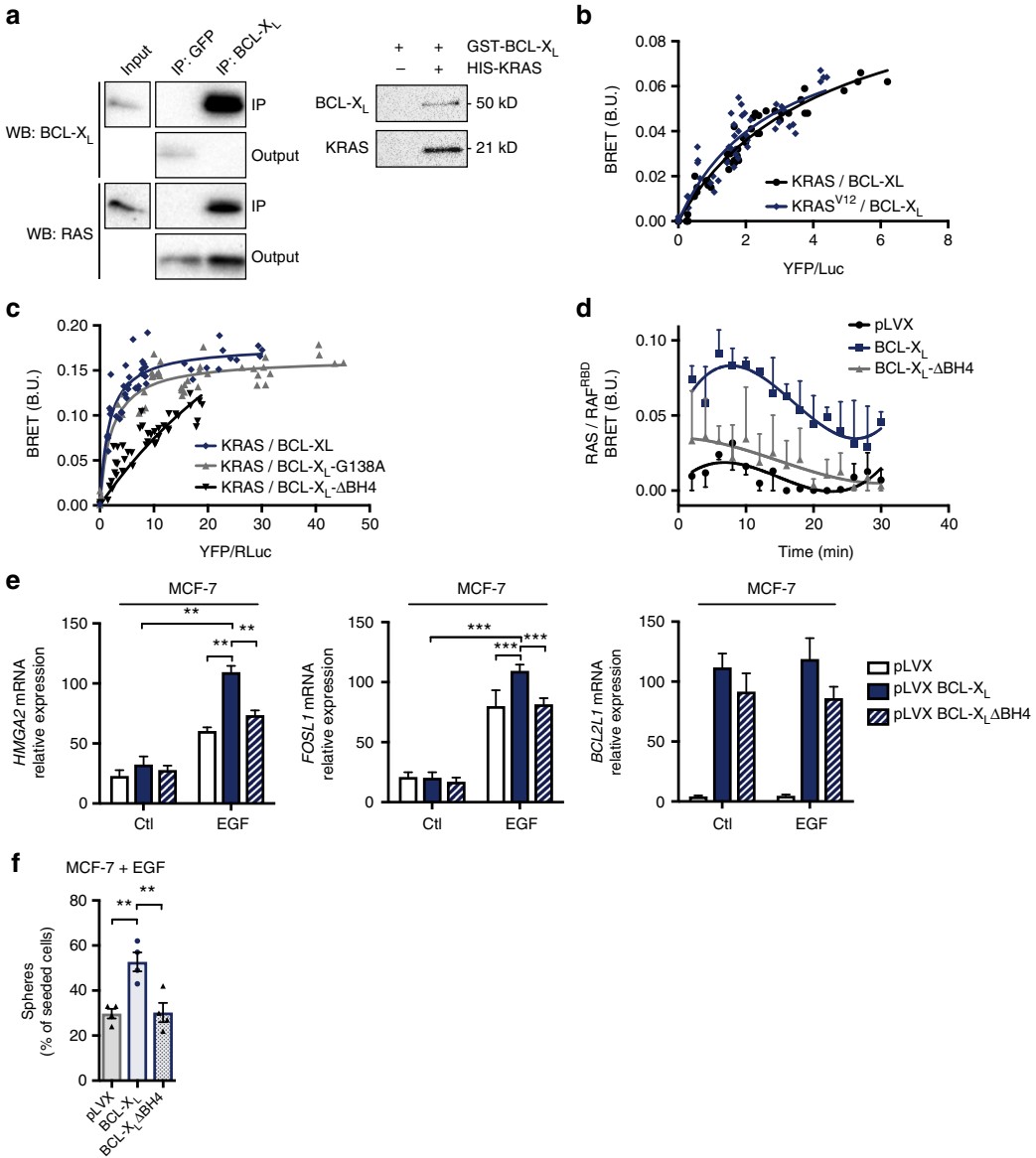

**Fig. 5** BCL-X$_L$ interacts with KRAS through its BH4 domain to favour its signalling. **a** Left: MCF10A KRASV12 lysate was used to perform immunoprecipitation with a control or an anti-BCL-X$_L$ antibody. Western blot immunodetection was done using anti pan-RAS or anti BCL-X$_L$ antibody. Right: Pulldown assays between His-tagged KRAS and GST-tagged BCL-X$_L$ were analysed by western blot immunodetection as in immunoprecipitation assay (see Methods for more details). **b, c** Interaction between KRAS and BCL-X$_L$ in MCF-7 cells was assessed by BRET saturation curve assays using increasing amount of plasmid encoding YFP-BCL-XL, YFP-BCL-XL-ΔBH4, or YFP-BCL-XL-G138A and a fixed amount of plasmid encoding Rluc-KRAS. BRET ratios were obtained for every YFP- BCL-X$_L$ plasmid concentration and plotted as a function of the ratio of total acceptor fluorescence to donor luminescence. Data were fitted using a nonlinear regression equation assuming a single binding site. Data presented are representative of three independent experiments. **d** RAS activity following EGF stimulation. BRET-based KRAS activity sensor (see Methods for details) was used to monitor the activity of KRAS after adding 200 nM EGF to cell medium at t = 0. Measurements were done in control cells (pLVX), BCL-X$_L$ overexpressing cells (BCL-X$_L$) and BCL-X$_L$-ΔBH4 overexpressing cells (ΔBH4). Data presented are representative of three independent experiments. **e** qPCR analysis of *HMGA2, FOSL1* and *BCL2L1* mRNA in MCF7 control cells (pLVX), BCL-X$_L$ overexpressing cells (BCL-X$_L$) and BCL-X$_L$ ΔBH4 overexpressing cells (ΔBH4) after overnight starvation and treatment with 20 ng ml$^{-1}$ EGF for 24 h. Mean and SEM of 3 independent experiments are represented as relative quantity of mRNA normalised to the mean of *RPLP0, RPS18* and *ACTB* relative expression (two-tailed unpaired t-test). **f** Percentage of sphere forming cells in bulk population of EGF-treated MCF-7 cells stably transfected with either a control plasmid (pLVX), a plasmid encoding BCL-X$_L$, or a plasmid encoding BCL-X$_L$-ΔBH4. Mean and SEM of 4 independent experiments are represented (two-tailed unpaired t-test)

## Discussion

In some cancers, accumulation of high BCL-X$_L$ expressing cells might emerge from a negative selection induced by cell autonomous pro-apoptotic oncogenic signals such as these resulting from enhanced MYC or decreased pRB expression. The situation appears to be different in basal-like breast cancers since BCL-X$_L$ expression is neither positively correlated to MYC expression nor negatively with that of pRB (Supplementary Fig. 5k). We found, instead, that BCL-X$_L$ expression levels correlate with those of KRAS and downstream RAF activation. This hints on a link between RAS activity, which is frequently high in basal-like breast cancers (despite rare activating mutations)[9], and BCL-X$_L$ expression whose expression is associated with therapeutic resistance in the same cancers. This correlation is mostly

**Table 2 Oligo sequences used for qPCR**

| | |
|---|---|
| ACTB | 5′-AGAAAATCTGGCACCACACC/CAGAGGCGTACAGGGATAGC-3′ |
| B2M | 5′-CGTGGCCTTAGCTGTGC/AATGTCGGATGGATGAAACC-3′ |
| BAK1 | 5′-GCCCACGGCAGAGAATGCCT/AGGGCCAGACGGTAGCCGAA-3′ |
| BAX | 5′-GCAACTTCAACTGGGGCCGGG/GATCCAGCCCAACAGCCGCTC-3′ |
| BBC3 | 5′-ACCTCAACGCACAGTACGA/GCACCTAATTGGGCTCCATC-3′ |
| BCL2 | 5′-CCTTCTTTGAGTTCGGTGGG/TCTTCAGAGACAGCCAGGAG-3′ |
| BCL2A1 | 5′-TGGATAAGGCAAAACGGAGGCTGG/CTTGTGGGCCACTGACTCTACCA-3′ |
| BCL2L1 | 5′-TTCAGTGACCTGACATCCCA/TCCACAAAAGTATCCCAGCC-3′ |
| BCL2L11 | 5′-GCCTTCAACCACTATCTCAG/TAAGCGTTAAACTCGTCTCC-3′ |
| BCL2L2 | 5′-TGGCCTACCTGGAGACGCGG/CAGTTCCCCTCCCGCAGACG-3′ |
| CCND1 | 5′-CAATGACCCCGCACGATTTC/CATGGAGGGCGGATTGGAA-3′ |
| CDH1 | 5′-GGGCGAGTGCCAACTGGACC/CCAGCGGCCCCTTCACAGTC-3′ |
| FN1 | 5′-AGAAGTGGTCCCTCGGCCCC/GGGTTACCAGTTGGGGAAGCTCG-3′ |
| FOSL1 | 5′-CAGGCGGAGACTGACAAACTG/TCCTTCCGGGATTTTGCAGAT-3′ |
| GAPDH | 5′-CAAAAGGGTCATCATCTCTGC/AGTTGTCATGGATGACCTTGG-3′ |
| HMGA2 | 5′-AGGCAGACCTAGGAAATGGC/CCAACTGCTGCTGAGGTAGA-3′ |
| HPRT1 | 5′-ATGCTGAGGATTTGGAAAGG/GATGTAATCCAGCAGGTCAGC-3′ |
| MCL1 | 5′-TCGGTACCTTCGGGAGCAGGC/CCCAGTTTGTTACGCCGTCGCT-3′ |
| PMAIP1 | 5′-CTCTGTAGCTGAGTGGGCG/CGGAAGTTCAGTTTGTCTCCA-3′ |
| RPLP0 | 5′-AACCCAGCTCTGGAGAAACT/CCCCTGGAGATTTTAGTGGT-3′ |
| RPS18 | 5′-ATCCCTGAAAAGTTCCAGCA/CCCTCTTGGTGAGGTCAATG-3′ |
| SNAI1 | 5′-GACCCCAGTGCCTCGACCACTA/CAGCAGGTGGGCCTGGTCGTA-3′ |
| VIM | 5′-GAGAACTTTGCCGTTGAAGC/TCCAGCAGCTTCCTGTAGGT-3′ |
| ZEB1 | 5′-TGGGAGGATGACAGAAAGGAAGGGC/TGCCTCTGGTCCTCTTCAGGTGC-3′ |
| ZEB2 | 5′-CACTATGGGGCCAGAAGCCACG/TGCTGACTGCATGACCATCGCG-3′ |

consistent with our mechanistic studies that provide evidence for a self-amplificatory process wherein RAS activity leads to induction of BCL-X$_L$ that in turn regulates RAS protein levels and signalling. In our assays BCL-X$_L$ induction and BIM repression were sensitive only to high concentrations of RAF or MEK inhibitors. This indicates that RAS targeting on its own might not be always sufficient to down-regulate BCL-X$_L$ and impact on cancer cell viability. The robustness of BCL-X$_L$ expression downstream of RAS activity is in agreement with BCL-X$_L$ described as a restrain to RAS targeting therapy and as one main target for synthetic lethal approaches using MEK inhibition[23]. This underscores the importance of the positive control BCL-X$_L$ exerts on RAS expression and signalling. It implies, moreover, that targeting this feedback itself may represent an interesting therapeutic approach impacting on some critical RAS signalling outcomes.

Attempts at controlling RAS activity have so far focused on the regulation of its GTP-binding and on post-translational modifications controlling its membrane association. Our data bring further support to the recently developed notion that investigating what potentiates or attenuates RAS beyond these processes is critical to understand the full oncogenic repertoire of RAS and to pharmacologically attack it[24]. Importantly, we define BCL-X$_L$ as one novel key regulator of RAS protein levels, functioning as a direct binding partner. We are currently investigating the regions involved in the binding interface(s) between KRAS and BCL-X$_L$ to understand the mechanistic basis and the exact consequences of BCL-X$_L$ binding on KRAS protein and its turnover.

Our study is also consistent with the idea that distinct biological outputs will result from differing RAS signal intensities. Indeed, we found that BCL-X$_L$ depletion and its subsequent intermediate effect on ERK phosphorylation does not affect RAS-driven cell proliferation, single cell motility or glycolysis (data not shown). On the other hand, it significantly prevents the induction of HMGA2 and FOSL1 and the acquisition of a CIC phenotype. This implies that induction of phenotypic plasticity and self-renewal by RAS activity will not be automatic but will happen when signalling is optimal thanks to the involvement of regulators such as BCL-X$_L$. This is in agreement with studies that

have described an enrichment in BCL-X$_L$ expression in subsets of breast cancer cells endowed with stem cell properties[25]. If a critical role has been assigned to BCL-X$_L$ in the viability of some embryonic or cancer stem cells[26, 27], we suggest here that BCL-X$_L$ can also play a direct active role in the biology of CIC through its ability to modulate RAS activity.

The non-apoptotic role for BCL-X$_L$ we describe here implies that its enhanced expression can be advantageous before impacting survival. Thus, apoptosis resistance of BCL-X$_L$ over-expressing cancer cells is not necessarily the directly selected trait. At the core of this exaptation process, we define a direct stabilising interaction between BCL-X$_L$ and KRAS regardless of its mutational status. A similar interaction was reported to allow cell survival upon accumulation of post-translationally modified KRAS at the mitochondria[28]. The oncogenic consequences on RAS signalling were not investigated in this study. Our observations also evoke BCL-2/HRAS interactions and their reported effects on de-differentiation of luminal breast cancer cells, by mechanisms that have not been totally described yet[29]. We provide evidence for a role of the KRAS/BCL-X$_L$ interaction in the transition towards a CIC state together with the conceptual framework and tools to unravel its molecular basis, its regulation and biological outputs in breast cancer cell populations and CIC subsets.

Targeting the anti-apoptotic function of BCL-X$_L$ to destruct chemoresistant cells remains problematic. Currently available compounds lack full efficiency in cancer cells while inducing dose-limiting thrombocytopenia. Our description of a functional dialog between RAS signalling and BCL-X$_L$ provides a novel insight into what drives the expansion of chemoresistant cells and offers the possibility to develop novel therapeutic strategies that can counteract the positive selection process that favours it.

## Methods

**Cell lines**. MCF10A Lxsn and KRAS$^{V12}$ cells were grown in DMEM-F12 (Gibco, Saint Aubin, France) supplemented with 5% Dubelcco Horse Serum (DHS) (Eurobio, Courtabeouf, France), glutamine 2 mM (Gibco), hydrocortisone 0.5 μg ml$^{-1}$ (Sigma-Aldrich), cholera toxin 100 ng ml$^{-1}$ (Sigma-Aldrich), insulin

10 µg ml$^{-1}$ (Sigma-Alcrich), HEPES 10 mM (Sigma-Aldrich) and 20 ng ml$^{-1}$ EGF (PeproTech). Experiments were performed using the same medium but with 2% DHS. When indicated, 20 ng ml$^{-1}$ EGF was added to MCF10A Lxsn cells. MCF-7 cells were obtained from ATCC and grown in RPMI 1640 (Gibco) supplemented with 10% Fetal Bovine Serum (FBS) (Biosera, Boussens, France) and glutamine 2 mM. Stable models of MCF-7 or MCF10 KRAS$^{V12}$ cells expressing pLVX (empty vector), pLVX BCL-X$_L$ or pLVX BCL-X$_L$ ΔBH4 were obtained by transfection with Lipofectamine 2000 ® according to the manufacturer protocol in the case of MCF-7 cell line or by lentivirus infection in the case of MCF10A cell line. Selection was performed with 500 ng. ml$^{-1}$ of puromycin. When specified lentivirus were used to transduce control lentivectors or sh-RNA (MOI 5). When specified we used QVD-OPh (R&D System, Minneapolis, MN, USA), ABT-737 (Sigma, St. Louis, MO, USA), L779450 (Santa Cruz Biotechnologies, Heidelberg, Germany), U0126 (InvivoGen, San Diego, USA) and CHX (Sigma, St. Louis, USA).

For the rescue experiment, MCF10A KRAS$^{V12}$ were infected with lentiviruses encoding for either a sh-RNA resistant BCL-X$_L$ coding sequence (the resulting lentivector was named pLVX BCL-X$_L$ for the sake of simplicity, mutated nucleotides in the shRNA resistant *BCL2L1* cDNA sequence are underlined: (60) 5′-AGGCTACTTCTGGAGTCAG-3′ (78)) or a control empty vector (pLVX). RNA interference was then performed by lentivirus infection using sh-RNA sequence targeting BCL-X$_L$ before carrying out sphere forming assays as described below.

For the CRISPR Cas9-induced BCL-X$_L$ knockout (KO), single guide RNA targeting Human BCL-X$_L$ was designed using the MIT CRISPR design tool (http://crispr.mit.edu/). The following guide sequence (5′-GCAGACAGCCCCGC GGTGAA-3′) was cloned in the plentiCRISPRV2 vector guide. Empty vector was used as a control. Cells were selected using 1 µg.ml$^{-1}$ puromycin and BCL-X$_L$ KO was confirmed by western blot.

**qPCR analysis**. Total RNAs were isolated using the Nucleonspin® RNA kit (Macherey-Nagel, Düren, Germany) according to the manufacturer protocol. Retrotranscription was performed using the Maxima First Strand cDNA Synthesis Kit (Fisher Scientific, Pittsburgh, PA, USA). mRNA expression was quantified by qPCR using EurobioGreen qPCR Mix Lo-Rox (Eurobio, Courtaboeuf, France) on qTOWER instrument (Eurobio, Courtaboeuf, France). Reaction was done in 10 µl final with 4 ng RNA equivalent of cDNA and 150 nM primers. Relative quantity of mRNA was estimated by Pfaffl method[30] and normalised on the mean relative quantity of three HKGs selected with GeNorm[31]. Primer sequences are listed in Table 2.

**Biochemical assays**. Immunoprecipitation assays were performed as follows: cells were cultured in 10 cm petri dishes and were collected and washed with PBS. Cell lysis was performed using ChIP buffer (SDS: 1% EDTA: 10 mM, Tris-Hcl ph 8,1: 50 mM (plus a cocktail of protease and phosphatase inhibitors)). and cellular suspensions were sonicated for 15 min thrice. 10 µl of anti-BCL-xL (Abcam) or 2 µl of anti-GFP antibody (Abcam) were used for 500 µg of cell extract to carry out immunoprecipitations that were performed as described in the PureProteome™ Protein G Magnetic Beads protocol (Millipore).

The pulldown protocol was adapted from the HisPur™ Ni-NTA Magnetic Beads procedure (Thermo scientific). Briefly, 1µg of His tagged KRAS (Abcam #ab96817) was mixed to 40 µl of Ni-NTA Magnetic beads in 400 µl of binding buffer (25 mM Tris●HCl, pH 7.2, 150 mM NaCl, 5 mM MgCl2, 1% NP-40, 20 mM Imidazol) for 60 min in an end-over-end rotator at room temperature. Beads were then collected on a magnetic stand and washed one time with binding buffer to remove unbound proteins. 0,05 µg of GST-tagged BCL-X$_L$ (Cliniscience #H00000598) in 400 µl of binding buffer was added to the beads and mixed overnight at 4 °C. Beads were then washed thrice in wash buffer (25 mM Tris-HCl, pH 7.2, 150 mM NaCl, 5 mM MgCl2, 1% NP-40, 50 mM Imidazol). Bound proteins were eluted using elution buffer (25 mM Tris●HCl, pH 7.2, 150 mM NaCl, 5 mM MgCl2, 1% NP-40, 200 mM Imidazol) and analysed by western blot immunodetection.

For western blotting, following SDS–PAGE, proteins were transferred to 0.45 µM PVDF membranes using Trans-Blot® Turbo™ Mini PVDF Transfer Packs (Bio-Rad) and a Trans-Blot® Turbo™ Transfer System Cell system (Bio-Rad). The membrane was then blocked in 3% BSA TBS 0.1% Tween 20 and incubated with primary antibody overnight at 4 °C. Blots were incubated with the appropriate secondary antibodies for 1 h at room temperature and visualized using the ChemiDoc XRS+ system (Bio-Rad). Primary antibodies used were anti-KRAS (Santa Cruz, sc-30), anti-BCL-X$_L$ (abcam, ab32370), anti-MCL-1 (Santa Cruz, sc-819), anti-BIM (Millipore, #AB17003), anti-p-ERK (Cell Signaling, 4370), anti-ERK (Cell Signaling, 9102) and anti-b-ACTIN (Millipore, MAB1501R). The full blots corresponding to portion of blots presented in the main figures are shown in Supplementary Fig. 8.

**Flow cytometry analysis**. Cells were harvested using trypsin, gentle agitation and washed with PBS. 300,000 cells were used for each staining. Membrane staining was performed on fresh cells using 15 ng of CD44-APC (559 942, BD Bioscience, Le Pont de Claix, France) antibody or isotypique control-APC (559 745, BD Bioscience) in PBS 0.5% BSA incubated 15 min in obscurity. Cells were washed in PBS BSA 0.5%, then fixed with 2% formaldehyde during 10 min at 37 °C. Cells were

washed with PBS 0.5% BSA and then permeabilised with 90% cold methanol during 30 min at 4 °C and then washed twice. Intracellular staining was performed with 2 ng of BCL-XL-A488 antibody (2767S, Cell Signaling Technologies, Leiden, The Netherlands) or isotypic control-A488 (4340S, Cell Signaling Technologies) incubated 1 h in obscurity. Cells were eventually washed in PBS 0.5% BSA and analysed by flow cytometry, using a Canto II cytometer (BD Bioscience) operated by DIVA software. At least 20,000 events were collected per sample.

**Sphere forming assay**. After filtration through 40 µm cell strainer (BD Bioscience, Le Pont de Claix, France), for each condition, 4 cells per well were seeded in 32 wells of 96 well ultra-low attachment plate (Corning, Avon, France) in MEBM bulletkit media (Lonza, Levallois-Perret, France) supplemented with B27 (Life Technologies, Saint Aubin, France). For each condition in each independent experiment, 128 cells were seeded. Each experiment was repeated at least 3 times. Mammosphere percentage of seeded cells was determined after 15 days incubation. On mammosphere figures, each dot represents the percentage of seeded cells that formed mammospheres in one independent assay.

**iTraq and mass spectrometry analysis**. Approximately $5 \times 10^6$ cells were lysed in 0.6 ml of 4% SDS and 0.1 M DTT in 0.1 M Tris-HCl, pH 7.6 and briefly sonicated. The samples were prepared and analysis done as previously described[17]. Briefly, proteins were extracted and digested with trypsin. The resulting peptides were labelled with iTraq reagents and fractionated by a 2D-OFFGEL approach. Two independent experiments were performed and for each protein detected in both experiments the geometric mean was calculated (See Supplementary Methods for more details).

**Mice**. Animal experiments were performed in accordance with the French regulations and approved by the local animal ethics committee (License No. CEEA.2012.84). MCF10A-KRAS$^{V12}$ cells were transduced using either control lentivectors or *BCL2L1*-targeting sh-RNA (MOI 5). Four days later cells were washed, harvested in PBS and mixed (50:50) in Matrigel (BD Biosciences) before injection in 9-week-old female nude mice (Swiss Nu/Nu, Charles River Laboratories). 7 mice were assayed for each group. We subcutaneously injected into the right flank of mice $5 \times 10^4$ cells in a final volume of 150 µl. Importantly, the number of transplanted-KRAS$^{V12}$ cells was determined by serial dilution assays as the minimal to inject in order to obtain 100% tumour uptake in less than 5 months (not shown). The presence of a visible or palpable tumour was then regularly monitored and tumour volume was measured using callipers during a period of 32 weeks. The animals were euthanized when tumour volume reached 2000 mm$^3$ or when signs of tumour necrosis were observed.

**Immunohistochemistry**. 104 patients diagnosed and treated at the ICO Cancer Center were collected between 1998–2007 among triple negative invasive breast carcinoma (ER/PR and Her2 negative). Representative formalin fixed tumours blocks were selected to establish tissue microarray for 88 patients. For 16 patients, whole tumour block was used as tissue microarray was defective.

Immunolabelling technique was performed by the Benchmark XT automated tissue staining system (Ventana Medical system) on 4 µm thick blocks section. Primary antibodies used were BCL-X$_L$ (BD Pharmingen, rabbit polyclonal 556361, CC1 short PH8.4, dilution 1/500), and p-ERK (Phospho-p44/42 MAPK (thr 202/ tyr 204) Cell Signaling, rabbit monoclonal, CC1 standard PH8.4, dilution 1/400)

For each staining, the H-Score was calculated as "intensity of staining" x "% of stained cells" where the "intensity of staining" was graded from 0 to 5 (0 none, 1 very weak, 2 weak, 3 intermediate, 4 strong and 5 very strong) and the "percentage of stained cells" estimated from number of tumours cells with cytoplasmic and/or nuclear staining (in case of p-ERK staining) or cytoplasmic staining (in case of BCL-X$_L$ staining).

**BRET**. RLuc expression plasmids were constructed by subcloning KRAS coding sequences into the pRLuc-C2 vector (BioSignal Packard). eYFP expression plasmids were constructed by subcloning BCL-X$_L$ (or derivatives thereof) coding sequences into the pEYFP-C1 vector (BD Biosciences). All constructs were sequenced before use. BRET saturation curves assays: at 24 h before transfection, cells were plated in 12-well plates. Cells were transfected with increasing amounts (50 to 1500 ng per well) of plasmids coding for a BRET acceptor (eYFP-BCL-X$_L$ and BCL-X$_L$ ΔBH4), and constant amounts (50 ng per well) of plasmid expressing the BRET donor RLuc-KRAS, using Lipofectamine 2000 (Life Technologies) according to the manufacturer's instructions. 24 h later, cells were collected and seeded in duplicates in 96-well white plates. 24 h later, cells were treated during 16 h. Prior to BRET measurement, cells were washed once with PBS. Coelenterazine H substrate (Interchim) was injected in plates, in PBS at a final molarity of 5 µM, and BRET was measured immediately and at 5 successive times. BRET was monitored using the lumino/fluorometer Mithras LB 940 (Berthold Technologies, France), allowing for the sequential integration of luminescence with two filter settings. The emission signal values obtained at 530 nm were divided by the emission signal values obtained at 485 nm. The BRET ratio was calculated by subtracting the BRET signal value obtained with co-expressed donor and acceptor by that obtained with the donor protein expressed alone. Data shown are

representative of at least three independent experiments. KRAS activity sensor: expression plasmid encoded eYFP-KRAS was co-transfected with donor plasmid pRluc-RAF-RBD (500 and 300 ng respectively per well of 12 wells plate). 24 h later, cells were collected and seeded in 96-well plates and allowed to adhere for 16 h before being starved from FBS for 24 h. EGF was added to the wells just before BRET measurements begun (200 ng ml$^{-1}$).

**Statistical analysis of TCGA data**. RPPA trimmed and RPPA subtype calls datasets established in the 2012 Breast cancer study[32] were downloaded from the TCGA data portal (https://tcga-data.nci.nih.gov/docs/publications/brca_2012). Correlation plots between proteins of interest for each breast cancer subtype and Pearson coefficient calculation were achieved using R program.

**Data availability**. The authors declare that the main data supporting the findings of this study are available within the article and its Supplementary Information files. Extra data are available from the corresponding author upon request.

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

## Acknowledgements

We thank members of the "Stress adaptation and tumor escape" laboratory for their support. We are particularly indebted to D. L. Maillet for his help and criticism. We thank Dr Y. Guillemin for his help and inspiration at the beginning of this work and P. East for proof reading this manuscript. We thank Dr R. Rimokh and Prof. G. Gillet for fruitful discussions. We thank Dr Ho Park for his generous gift of the MCF10-A cell lines and Dr O. Micheau for his gift of pLVX (BCL-X$_L$). We thank C. Couriaud for her technical help in the preparation of lentivirus particles. We benefited from invaluable technical support from the Cytometry Core facility (CytoCell) of the Federative Research Structure François Bonamy (Nantes). Dr S. de Carné Trécesson was supported by Institut National du Cancer, Dr J. Pécot by Ministère de la Recherche et de l'Enseignement Supérieur and Dr A. Basseville by Fondation de France. This work was supported by Canceropole Grand Ouest (CIC project 2010–2012, MATURE project 2017–18), ARC (R15083NN), Fondation de France (2015) and INCA PLBio (R12134NN) to PJ, Ligue Grand Ouest (R13137) to S.C.T. and P.P.J and Fondation de France (2017) to A.B.

## Author contributions

S.C.T., F.S., A.B., A.-C.B., J. P., M.B., K.A.S., S.B.N., I.V., O.C., C.G., F.G., and P.P.J. conducted experiments. S.C.T., J.L., A.L., S.B.N., M.C., F.G., and P.P.J. designed the experiments. S.C.T., F.S., A.B., A-C.B., K.A.S, S.B.N., I.V., O.C., C.G., F.G., and P.P.J. analyzed the data. S.C.T., F.G., and P.P.J. wrote the paper. S.C.T., A.B. and P.P.J. obtained funding. P.P.J. conceived the study and supervised it.

## Additional information

**Competing interests:** The authors declare no competing financial interests.

