## [Peer Review File · Nature Communications]

Reviewers' comments:

Reviewer #1 (Remarks to the Author):

This paper describes a novel interaction of RAS and Bcl-XL which modulates Ras activity and promotes signaling towards cancer stemness. This finding is novel and very interesting to the research community. I particularly liked the data on Bh4 domain deleted mutant (Figure 5d-f) which provide a lead to develop novel inhibitors of Bcl-XL.

However, while the findings presented appear robust it is unclear to me how the binding of Bcl-XL modulates Ras activity. The authors state in their discussion that Bcl-XL stabilizes Ras. Did they mean that the half-life of Ras is increased by Bcl-XL? I am not sure that the current paper presents sufficient evidence for this stabilization. In Fig S5a treatment with the translation inhibitor CHX is shown to reduce KRAS protein levels more prominently in the absence of Bcl-XL but is this sufficient evidence to say that Ras is stabilized by Bcl-XL? Would overexpression of Bcl-XL result in increased Ras activity / stability?

Also, they state that in order to induce a CIC phenotype Ras signaling needs to be optimal which is achieved by Bcl-XL. However, it is not clear to me how Bcl-XL can modulate Ras activity selectively and increase expression of some target genes and not others. What is the mechanistic consequence of Bcl-XL binding to Ras, and how does this modulate transcription induction? It would improve the impact of this manuscript if the authors could address these questions.

Minor points:

Supp Fig 1d : Can the authors justify why they have used Anoikis as a model for "anti-apoptotic traits"?

Fig 1d top: This Figure is not very well presented. Maybe use different symbols/ lines for IgG and Bcl-XL to not give the impression that there are two populations (Bcl-XL high and low). Bottom: Include geographic mean or median of the different subpopulations to better compare control and RAS cells. As this appears to be the starting point for the following investigations these data could be displayed in a clearer way.

Supp Fig1e is not currently linked with main text, maybe include in line 88/89

Fig 2b: In this western blot a loading control missing

Fig 2c: maybe include Lxsn untreated cells as comparison (currently included in SuppFig 1b)

Fig 2e: Quantify the percentage of cells with low CD44 upon BclX shRNA.

Fig S5g: Include Pear. value as in other correlation analysis.

Line 194: can you show the mass spec data for the BclXL IP?

Line 202/203: show the data on BclXL mutations and ABT737 to put the data provided in Fig 5c into perspective.

Reviewer #2 (Remarks to the Author):

In this very interesting paper the authors build a case for a functional interaction between Bcl-XL and KRAS in cancer cell stemness. The observation that Bcl-XL expression is selectively advantageous to cancer cell populations even in the absence of pro-apoptotic pressure is very interesting, novel and of interest to the field. Importantly they identify that the BH4 region of Bcl-XL rather than its BH3 binding region is required for the interaction. The comparative proteomic analysis and functional assays indicate that the interaction with Bcl-XL is critical for RAS-induced expression of stemness regulators and maintenance of a cancer initiating cell (CIC) phenotype. This data is clearly presented and the proteomic analysis and expression of stemness genes is convincing. The findings would be of significant general interest to both the cancer and apoptosis communities if the cellular data presented were equally compelling. Although, the data presented in the paper are very striking, very little primary cell data is provided making it difficult to understand the summary histograms.

Primary data should be shown for sphere forming assays in Figures 2 and S3. Histograms do not

make it clear that caspase inhibition did not rescue mammosphere formation. Furthermore it is difficult to evaluate whether "no effect on mammosphere formation was detected upon BH3 mimetic (ABT-737) treatment (Figure S3b and c)." from the data in the figure. What else was measured? How was a sphere defined in these assays? In the various experiments were there changes in the size or morphology of spheres? What number of wells (4 cells/well were plated to generate spheres) were prepared for each condition? How many independent experiments? In Sig. S1b it looks like four wells in a single experiment. What about the formation of acini? It has been reported that MCF10A LXS cells generate spherical acini structures with a hollow lumen. What is the effect of the two genes alone and in combination on the formation of acini? Importantly, the sphere forming assays are based on growth in minimal medium on a non-adherent substrate. How do the authors rule out a role for resistance to anoikis? If Bcl-XL is preventing anoikis then one would not see differences in cell death or resistance once mammospheres form but would see a pronounced effect on formation of mammospheres. They may well have controlled for this but I was unable to discern how it was approached in the manuscript.

Specific points for consideration:

To demonstrate that shRNA Bcl-XL effects are on target it is necessary to express a shRNA resistant mRNA for Bcl-XL in the cell line and show that this abrogates the effect of the shRNA. This control is essential for all critical data reported.

I don't follow the argument for BCL-XL depletion leading to decreased phosphorylation of ERK (p-ERK) in MCF10A KRASV12 cells and in EGF-treated cells (Figure 3c and 3d). In Figure 3c there is not a good correlation between Bcl-XL knockdown and p-ERK and in Figure 3d the exposure is not sufficient to judge whether or not there is a correlation. Assuming the same shRNAs are used in S5A (they are named differently) there isn't a good correlation between the knockdown activity and RAS protein turnover either.

The BRET studies suggesting KRAS binding to Bcl-XL is increased with an intact BH4 region are very interesting. However, the control experiments demonstrating that BRET signals between BCL-XL and KRAS were not inhibited by mutations in BCL-XL BH3 binding region including, demonstrating the BRET response with an authentic BH3-protein, are essential to interpreting this data but are not shown. Similarly resistance of the BRET signal due to KRAS Bcl-XL interaction to ABT-737 should be shown along with a control in which the drug displaces a BH3-protein in the same assay. These studies would be strongly supported by a demonstration using purified proteins that the interaction is direct. Such investigations should be straightforward as both proteins have been purified by multiple groups in the past.

I was unable to follow the reverse-phase protein array analysis as presented in the paper. I could not find anything in the methods sections that explained how these results were generated. The paragraph in question is lines 156-162 on page 5. The term "public data" is not very informative. The correlations reported are very strong therefore I thought the trends should be obvious from TCGA data available via Bioportal. However, it took me a long time and various trials before I figured out what the authors did and was able to reproduce their data. Please explain this more fully for other readers interested in the results you obtained.

The immunochemistry analyses were also confusing. While there is a very small positive correlation it appears that it is measured against a subjective assignment of H-score. No primary data are shown and although the positive slope is statistically significant the biological significance of such a small slope based on subjective assignment of score is uncertain. How certain is this data? Have I missed something in how this was assessed?

Response to reviewers' comments

(Reviewers' original comments in italic, authors' responses in non-italic.)

Reviewer #1 (Remarks to the Author):

This paper describes a novel interaction of RAS and Bcl-XL which modulates RAS activity and promotes signaling towards cancer stemness. This finding is novel and very interesting to the research community. I particularly liked the data on Bh4 domain deleted mutant (Figure 5d-f) which provide a lead to develop novel inhibitors of Bcl-XL. However, while the findings presented appear robust it is unclear to me how the binding of Bcl-XL modulates RAS activity.

Thank you for your evaluation of this manuscript. As described below, we have performed additional experiments to address the concerns you raised and we rewrote certain parts of the manuscript to clarify our message and discuss it better.

The authors state in their discussion that Bcl-XL stabilizes RAS. Did they mean that the half-life of RAS is increased by Bcl-XL? I am not sure that the current paper presents sufficient evidence for this stabilization. In Fig S5a treatment with the translation inhibitor CHX is shown to reduce KRAS protein levels more prominently in the absence of Bcl-XL but is this sufficient evidence to say that RAS is stabilized by Bcl-XL? Would overexpression of Bcl-XL result in increased RAS activity / stability?

Thank you for pointing out that writing that "BCL-XL stabilizes RAS" is an overstatement given currently available data. We observed by an assumption-free proteomic approach that BCL-XL down regulation by shRNA (Figure 2b) led to a reduction of KRAS protein levels and, most importantly, we confirmed this effect in this new version of the manuscript after knock out of *BCL2L1* by a CRISPR/Cas9 approach (this is shown in Figure S5a). In addition, in publicly available expression data from human breast cancers, we found a positive correlation between BCL-XL and KRAS protein expression (Figure 5). We propose in this manuscript that this effect ensues from an effect of BCL-XL on KRAS protein levels because BCL-XL down regulation enhances the inhibitory effects of cycloheximide on KRAS protein expression (as previously shown, Figure S5a) but also because it has no detectable effect on KRAS mRNA expression (these negative data, not shown, were not mentioned in the initial version of the manuscript, and it is now).

Regarding overexpression of BCL-xL, we show that it results in increased RAS activity over time. To establish this, we performed live cell measures of RAS binding to the RAS binding domain of RAF in cells overexpressing or not BCL-xL and stimulated them with EGF (Figure 5d). Overexpression of wild type BCL-xL, but not of the BH4 deleted variant, enhances EGF induction of RAS activity and it prolongs it overtime. Moreover, Figure 5e shows that wild type BCL-xL, but not the BH4 deleted variant, favours the induction of genes that are critically dependent on RAS signalling for their expression (as established in Figure 4). Finally, overexpressed BCL-xL (provided its BH4 domain is present) favours mammosphere formation (Figure 5f).

We rewrote the text at different places (essentially in pages 5-6 and in the Discussion) to mention an effect of BCL-XL on KRAS "protein levels" instead of "stability". It should be noted that multiple mechanisms activating or attenuating RAS activity in addition to those relying on guanine nucleotide exchange factors, GTPases-activating factors and modulators of membrane binding have been reported. Among these, there is a growing body of

evidence showing that mechanisms regulating KRAS protein turnover have a critical role to play (Pfleger, *Science Sign.*, 2011).

Also, they state that in order to induce a CIC phenotype RAS signaling needs to be optimal which is achieved by Bcl-XL. However, it is not clear to me how Bcl-XL can modulate RAS activity selectively and increase expression of some target genes and not others. What is the mechanistic consequence of Bcl-XL binding to RAS, and how does this modulate as transcription induction? It would improve the impact of this manuscript if the authors could address these questions.

We think our study contributes to the notion that the organisation of the RAS/MAPK pathway allows apparently subtle changes in stimulation to be converted into very different biological phenotypes, and that BCL-xL contributes to this. The RAS/MAPK pathway is subject to numerous feedbacks and it can filter inputs and transmits them into appropriate outputs by encoding not only intensity but also dynamics of stimuli (see Avraham and Yarden, *Nature Rev. Mol. Cell. Biol.*, 2011 for a review). Most relevantly here, it was proposed that decoding of RAS/MAPK amplitude and duration occurs at the level of gene transcription (Blüthgen *et al.*, 2009, *FEBS Journal*), with the expressions of some genes being particularly sensitive to such integration (Mackeigan *et al.*, 2005, *Mol Cell Biol*). We think that our manuscript brings further support to this notion by showing that fine regulation of KRAS protein abundance and of its downstream signalling, by BCL-XL, impacts on the expression of selective gene sets and consequently the induction of specific phenotypes. This is experimentally supported by: i) the observation that *HMGA2* expression, which we identified as being BCL-XL-dependent, is significantly affected by a mild inhibition of RAS/MAPK signalling (as opposed to that of *BCL2L1* itself or *CCND1*) (Figures 4a-b and S6a-b); ii) *FOSL1* expression, previously established as requiring a high MAPK amplitude for its induction (Mackeigan *et al.*, 2005, *Mol Cell Biol*), is BCL-XL-dependent (Figure 4c). To insist on this point, we now show in a new Figure 4 and Figure S6 the effects of a dose-range of Raf and MEK inhibitors (respectively) not only on mRNA expressions but also on phosphorylation of ERK. This new figure illustrates how small decreases in Raf or MEK activities (directly targeted in this experimental setting) lead to detectable changes in the mRNA expression of some genes, while changes in expression of other genes, or even in ERK phosphorylation appear to be less detectable. This is a critical experiment since it allows us to discriminate genes whose expression is highly sensitive to changes in RAS signalling (*HMGA2*, *FOSL1*) from these that are not (*CCND1*, *BCL2L1*). The fact that BCL-xL function finely modulates gene transcription downstream of RAS by its non-canonical activity is established by the fact that *HMGA2* and *FOSL1* expressions are favoured by overexpressed wild type BCL-xL, but not by the BH4 deleted variant (Figure 5e).

Even though the fact that regulation of KRAS abundance will impact on signalling output is consensual, the exact mechanism(s) involved are still subject to debates. Proteasomal degradation of poly-ubiquitinated RAS proteins (including KRAS) was shown to decrease their stability (Kim *et al.*, *J Cell Science*, 2009) but KRAS was also shown to be activated by non-degradative ubiquitination (Sasaki *et al.*, *Sci. Signal.*, 2011). Other groups have established that KRAS intracellular trafficking is such that this protein is prone to lysosomal degradation, by a process that erodes signalling (Lu *et al.* *JCB*, 2009). We provide evidence in this new version of the manuscript that the latter process is involved in our case. Indeed, Bafilomycin A1, an inhibitor of vacuolar-type H⁺-ATPase that prevents acidification and protein degradation in lysosomes, and chloroquin, another lysosomal inhibitor, both enhance

KRAS protein levels in BCL-xL knock out cells. In contrast, MG132, a proteasome inhibitor, has no detectable effect (Figure S5c). We feel that understanding how exactly BCL-XL controls KRAS lysosomal degradation requires a completely novel set of experiments that is beyond the scope of this study, which is itself focused on the description of the biological consequence of the KRAS/BCL-XL interplay in transformed mammary epithelial cells.

Minor points

Supp Fig 1d : Can the authors justify why they have used Anoikis as a model for “anti-apoptotic traits”?

Thank you for this comment. We have now clarified the choice of anoikis as one model for anti-apoptotic traits. Due to the limitation of space, we refer to anoikis in the main text but only provide explanations in Supplementary Information. Briefly, we addressed anchorage-dependent cell death in KRASV12-transformed cells because we had previously observed their enhanced ability to grow in mammosphere conditions, and we inferred that this feature requires resistance to anoikis. As shown in Figure S1d, KRASV12 cells were indeed resistant to anoikis compared to the parental cells, but this does not solely rely on BCL-XL expression (as sh-BCL-XL proved insufficient in itself to restore sensitivity). Of note, “anti-apoptotic” traits upon expression of KRASV12 were confirmed by the increased resistance of corresponding cells to death induction by a promiscuous pro-apoptotic compound (staurosporine, data not shown but mentioned in the Supplementary data) and by BH3-profiling assays (Figure S3e).

Fig 1d top: This Figure is not very well presented. Maybe use different symbols/ lines for IgG and Bcl-XL to not give the impression that there are two populations (Bcl-XL high and low). Bottom: Include geographic mean or median of the different subpopulations to better compare control and RAS cells. As this appears to be the starting point for the following investigations these data could be displayed in a clearer way.

Thank you for this suggestion. We reformatted this Figure as asked. This new version should be clearer. It provides, both in EGF treated LXS cells and in KRASV12 cells CD44 mean of expression in subpopulations of cells classified based on their BCL-xL expression.

Supp Fig1e is not currently linked with main text, maybe include in line 88/89

We have now included Figure S1e in the main text where you suggested.

Fig 2b: In this western blot a loading control missing

Thank you for highlighting this point. We have now added a loading control in addition to a western blot showing that total ERK levels are left unchanged by EGF treatment.

Fig 2c: maybe include Lxsn untreated cells as comparison (currently included in SuppFig 1b). incorporate data neg

Thank you for this suggestion. We have now added the condition of untreated Lxsn in Figure 2c and updated the legend accordingly.

Fig 2e: Quantify the percentage of cells with low CD44 upon BclX shRNA.

Thank you for your recommendation. You will find Figure 2e updated with the percentage of CD44low cells in control or sh-BCL-XL treated cells.

Fig S5g: Include Pear. value as in other correlation analysis.

Thank you for pointing this out. We have now included Pearson values in Figure S5.

Line 194: can you show the massspec data for the BclXL IP?

As asked, the new Supplementary Table 2 shows the list of proteins that were specifically identified after IP-BCL-XL in KRASV12-transformed cells. This gross mass spectrometric analysis of protein lysates from the cells of interest after IP-BCL-XL was performed to have a hint of candidate binding partners. Importantly, BCL-XL itself was systematically identified in these immunoprecipitations. We performed negative control experiments in parallel using lysates from the parental cells poorly expressing BCL-XL. BCL-XL was never identified in these immunoprecipitations but some other proteins were. We dubbed them as non-specific and took them off the short list of candidates shown in Supplementary Table 2.

Line 202/203: show the data on BclXL mutations and ABT737 to put the data provided in Fig 5c into perspective.

Thank you for your comment on this. We have now added two sets of data supporting the fact that the BH3-binding site of BCL-XL is not involved in its interaction with KRAS in a new Figure S7. Firstly, we investigated BRET signals between KRAS and a variant of BCL-XL that carries a point mutation affecting its binding to pro-apoptotic counterparts. We found these signals to be comparable to those measured using wild type BCL-XL (Figure 5c). Secondly, we now show that BRET signals between KRAS and BCL-XL do not decrease upon treatment with BH3 mimetics (i.e. ABT-737 or WEHI-539). In contrast, these compounds inhibit BRET signals between BCL-XL and pro-apoptotic tBID in a dose dependent manner (Figure S7).

Reviewer #2 (Remarks to the Author):

In this very interesting paper the authors build a case for a functional interaction between Bcl-XL and KRAS in cancer cell stemness. The observation that Bcl-XL expression is selectively advantageous to cancer cell populations even in the absence of pro-apoptotic pressure is very interesting, novel and of interest to the field. Importantly they identify that the BH4 region of Bcl-XL rather than its BH3 binding region is required for the interaction. The comparative proteomic analysis and functional assays indicate that the interaction with Bcl-XL is critical for RAS-induced expression of stemness regulators and maintenance of a cancer initiating cell (CIC) phenotype. This data is clearly presented and the proteomic analysis and expression of stemness genes is convincing. The findings would be of significant general interest to both the cancer and apoptosis communities if the cellular data presented were equally compelling. Although, the data presented in the paper are very striking, very little primary cell data is provided making it difficult to understand the summary histograms.

Thank you for your positive comments on our manuscript. We carried out additional experiments to explain further the mechanistic behind RAS/BCL-XL interaction. We have also brought clarification in the main text that we hope you will find now more straightforward.

Primary data should be shown for sphere forming assays in Figures 2 and S3. Histograms do not make it clear that caspase inhibition did not rescue mammosphere formation.

Furthermore it is difficult to evaluate whether “ no effect on mammosphere formation was detected upon BH3 mimetic (ABT-737) treatment (Figure S3b and c).” from the data in the figure. What else was measured? How was a sphere defined in these assays? In the various experiments were there changes in the size or morphology of spheres? What number of wells (4 cells/well were plated to generate spheres) were prepared for each condition? How many independent experiments? In Sig. S1b it looks like four wells in a single experiment.

We apologise for any confusion caused by the wording relating to the mammosphere experiment description. Thank you for pointing that out. You will find a detailed description in Supplementary Information of how exactly we evaluated the percentage of cells capable of forming a mammosphere in a given population of cells. We have also modified Figure 2c, 5f, S1b, S3b and S3c to resolve their ambiguity. We now show the mean of the indicated number of experiments (histograms) as well as the primary value (dots represent the percentage of cells capable of forming a mammosphere in the indicated conditions evaluated in an independent assay). We have added the statistical test results on the figure so that the lack of effect of ABT-737 treatment on sphere formation is more obvious, as well as the failure of Q-VD OPH treatment to restore sphere formation after sh-BCL-XL (Figure S3).

Regarding your question on sphere morphology, we noticed that the spheres formed after 15 days were of different sizes. Nevertheless, sh-BCL-XL affected the total number of spheres and not the size or the morphology of them specifically. Morphologically, we noticed that EGF-induced spheres were more spherical than KRASV12-induced spheres, which looked less compact and less organised with no defined basal membrane. We added a picture of representative EGF-induced and KRASV12-induced spheres (Please see Figure S1f). We think that this can be explained by the dedifferentiation occurring upon RAS-induced transformation of the cells used in this study.

What about the formation of acini? It has been reported that MCF10A LXS cells generate spherical acini structures with a hollow lumen. What is the effect of the two genes alone and in combination on the formation of acini?

We assume that you refer to acini formation assays. These are technically challenging assays where the ability of cells to form polarised structures upon growth in 3D in the presence of Matrigel is measured to evaluate the acquisition of epithelial traits (i.e. cell–cell adhesion, planar and apical–basal polarity and lack of motility). These assays are different from the mammary formation assays we performed here, that are devoid of Matrigel and that are designed to investigate the ability of each cell to give rise to a clone in minimal media and in the absence of adhesion. The latter are thus designed to enumerate CIC's in a cell population, being therefore the ideal ones to investigate the role of BCL-XL in cancer cell stemness.

We suspect that KRASV12-transformed MCF10A cells would not form acini. Firstly, as mentioned above, the mammospheres formed in KRASV12-transformed MCF10A were not spherical and not organised with a basal membrane like acini would be. This is somehow consistent with our demonstration that MCF10A cells lost their epithelial phenotype to display a sustained mesenchymal phenotype upon KRASV12 expression (Figure S1a and S1e). Secondly, it has been shown that the expression of oncogenes in MCF10A cells depolarised cells in spheroids leading to disrupt the morphogenic process of acini formation (see in Debnath *et al.*, 2003, Methods). For these reasons, we think that KRASV12-transformed MCF10A cells are unlikely to generate acini structures in semisolid media but

would instead develop in unorganised structures, due to their reported dedifferentiated state and resistance to anoikis, as discussed below.

Importantly, the sphere forming assays are based on growth in minimal medium on a non-adherent substrate. How do the authors rule out a role for resistance to anoikis? If Bcl-XL is preventing anoikis then one would not see differences in cell death or resistance once mammospheres form but would see a pronounced effect on formation of mammospheres. They may well have controlled for this but I was unable to discern how it was approached in the manuscript.

Thank you for pointing out the lack of information about this important part of the manuscript. We have modified the text, in the Supplementary Information, which describes the Figure S1d where we directly addressed resistance to anoikis, and showed that KRASV12 transformation promotes it in MCF10A cells. This resistance to anoikis is not solely dependent on BCL-XL expression as sh-BCL-XL is insufficient in itself to restore any detectable sensitivity to anoikis in KRASV12-transformed cells (Figure S1d). Going back to the above question about acini formation, anoikis is indeed one cell death process that was reported to contribute to the formation of a hollow lumen under these conditions. We thus infer from our data that KRASV12-transformed cells (expressing BCL-XL or not) would form filled 3D structures.

Specific points for consideration:

To demonstrate that shRNA Bcl-XL effects are on target it is necessary to express a shRNA resistant mRNA for Bcl-XL in the cell line and show that this abrogates the effect of the shRNA. This control is essential for all critical data reported.

Thank you for highlighting this important point. To ensure that the effects of shRNA were due to on-target effects, the initial experiments describing the role of BCL-XL were performed with three different sh-RNA sequences targeting BCL-XL (named sh-BCL-XL#1, sh-BCL-XL#2 and sh-BCL-XL#3 in the figures). Despite different efficacies to knock down BCL-XL, sh-RNA#2 and #3 recapitulated the effects observed with sh-BCL-XL#1 (Figure 2). We performed the rescue experiment you rightfully asked by treating with sh-BCL-XL#1 KRASV12-transformed cells after their infection with a lentivirus encoding for a sh-RNA resistant variant of BCL-XL cDNA. As shown in Figure S2d, these cells, in contrast to control cells, did not decrease their amount of mammosphere forming cells.

To further nail down this point, we used another model where *BCL2L1* (BCL-XL gene) was knocked out using CRISPR/Cas9 in KRASV12 cells. Using this model, we recapitulated the effect on sphere formation observed after sh-BCL-XL (Figure S2e)

As a whole, we feel that our data do support that the observations made after BCL-XL knockdown by RNA interference are due to an on target effect and a decrease in BCL-XL protein expression.

I don't follow the argument for BCL-XL depletion leading to decreased phosphorylation of ERK (p-ERK) in MCF10A KRASV12 cells and in EGF-treated cells (Figure 3c and 3d). In Figure 3c there is not a good correlation between Bcl-XL knockdown and p-ERK and in Figure 3d the exposure is not sufficient to judge whether or not there is a correlation. Assuming the same shRNAs are used in S5A (they are named differently) there isn't a good correlation between the knockdown activity and RAS protein turnover either.

Thank you for raising this point. We acknowledge that the modulation of ERK phosphorylation does not perfectly follow levels of BCL-XL remaining after shRNA treatment in Figure 3. Across all the shRNA experiments we performed, we always observed a better correlation between BCL-XL extinction and downstream expression (id est HMGA2) (Figure 3g) than between BCL-XL extinction and ERK phosphorylation decrease. MAPK activity has been shown to follow different and heterogeneous dynamics of activation leading to “waves” of ERK phosphorylation (Ryu *et al.*, 2015, Mol Syst Biology). Moreover, RAS induces expression of MAPK inhibitors like DUSP phosphatases which dampen the RAS-MAPK pathway signalling - ERK phosphorylation in particular - in a negative feedback loop (Sweet-Cordero *et al.*, 2005, Nat Genet ; Courtois-Cox *et al.*, 2006, Cancer Cell). These complex links between upstream RAS and downstream phosphorylation of ERK could explain the different levels of p-ERK we detected following differing sh-BCL-XL depletions. However, our data (further illustrated by the new Figure 4) hint that p-ERK does not need to be strongly affected to significantly impact on the expression on stemness regulators. Such a fine tuning is consistent with our whole data supporting the idea that BCL-XL contributes to a full activation of RAS in order to maintain CIC features but that it is not required for other RAS-dependent processes (e.g. cell survival, proliferation) that require less optimal signalling.

Thank you for pointing out the discrepancy in sh-RNA names across Figures. We have now homogenised legends in all Figures.

The BRET studies suggesting KRAS binding to Bcl-XL is increased with an intact BH4 region are very interesting. However, the control experiments demonstrating that BRET signals between BCL-XL and KRAS were not inhibited by mutations in BCL-XL BH3 binding region including, demonstrating the BRET response with an authentic BH3-protein, are essential to interpreting this data but are not shown. Similarly resistance of the BRET signal due to KRAS Bcl-XL interaction to ABT-737 should be shown along with a control in which the drug displaces a BH3-protein in the same assay.

We have now performed the experiments you suggested. We investigated BRET signals between KRAS and a variant BCL-XL that carries a point mutation affecting its binding to pro-apoptotic counterparts. We show they are not significantly different from these measured using wild type BCL-XL (Figure 5c). Moreover, we have investigated the influence of BH3 mimetics (that inhibit BRET signals between BCL-XL and pro-apoptotic tBID in a dose dependent manner) on KRAS/BCL-XL BRET signals and found no detectable effects of these compounds

These studies would be strongly supported by a demonstration using purified proteins that the interaction is direct. Such investigations should be straightforward as both proteins have been purified by multiple groups in the past.

Following your recommendation, we carried out an assay with recombinant proteins which shows pull down of GST-tagged BCL-xL when using HIS-tagged KRAS as a bait, and further enforces the notion that the two proteins interact directly (Figure 5a, right).

I was unable to follow the reverse-phase protein array analysis as presented in the paper. I could not find anything in the methods sections that explained how these results were generated. The paragraph in question is lines 156-162 on page 5. The term “public data” is not very informative. The correlations reported are very strong therefore I thought the trends

should be obvious from TCGA data available via Bioportal. However, it took me a long time and various trials before I figured out what the authors did and was able to reproduce their data. Please explain this more fully for other readers interested in the results you obtained.

Thank you for pointing out this oversight on our part. We have updated the Method section with the description of the statistical analysis and the cohorts used. Please note that we have not used cBioportal online tools but downloaded corresponding data and processed them in R. We have also specified in the main text as well as in the figure legends.

The immunochemistry analyses were also confusing. While there is a very small positive correlation it appears that it is measured against a subjective assignment of H-score. No primary data are shown and although the positive slope is statistically significant the biological significance of such a small slope based on subjective assignment of score is uncertain. How certain is this data? Have I missed something in how this was assessed?

We acknowledge that the slope of the dot plot in Figure S5g is relatively small. However, as you mentioned the positive correlation is significant, indicating that high BCL-xL expression tends to associate with high phosphorylation of ERK. We thus decided to add this graph to the set of supplementary figures as it supports - with additional data on a new cohort- the link between BCL-XL expression and RAS pathway activation found in the TCGA cohort of basal-like breast cancer.

Regarding the comment on H-Score, the way it has been assessed is described in the method: we calculated H-Score as “intensity of staining” x “% of stained cells” where the “intensity of staining” was graded from 0 to 5 (0 none, 1 very weak, 2 weak, 3 intermediate, 4 strong and 5 very strong) and the “percentage of stained cells” estimated from number of tumours cells with cytoplasmic and/or nuclear staining (in case of p-ERK staining) or cytoplasmic staining (in case of BCL-X_L staining). This method is commonly used in immunohistochemistry. It can be further evaluated by including the sum of individual H-scores for each intensity level seen. In that case the formula to use is [1 × (% cells 1+) + 2 × (% cells 2+) + 3 × (% cells 3+)] but our pathologist observed a homogenous intensity of staining which was evaluated from 0 to 5. Thank you for suggesting to show the primary data. We have added representative pictures of nuclear and cytoplasmic p-ERK and BCL-XL staining in Figure S 5i.

REVIEWERS' COMMENTS:

Reviewer #1 (Remarks to the Author):

I believe that the manuscript has been improved during the review process and is now acceptable for publication

Reviewer #2 (Remarks to the Author):

The authors have made a series of improvements to the manuscript and have addressed all of the queries raised by the referees adequately.

The information now provided for the coimmunoprecipitation assays reveals that non-ionic detergent was used. These detergents have been shown to promote artificial interactions between Bcl-2 family proteins in the past. The authors should consider commenting on this. They also now show that the pull-down reactions for mass spec were performed using SDS. I am not aware of published data on the effects of SDS on this class of proteins.

Reviewer #1 (Remarks to the Author):

I believe that the manuscript has been improved during the review process and is now acceptable for publication

We thank this reviewer for his positive appreciation of our revision.

Reviewer #2 (Remarks to the Author):

The authors have made a series of improvements to the manuscript and have addressed all of the queries raised by the referees adequately.

We thank this reviewer for this positive comment.

The information now provided for the coimmunoprecipitation assays reveals that non-ionic detergent was used. These detergents have been shown to promote artificial interactions between Bcl-2 family proteins in the past. The authors should consider commenting on this. They also now show that the pull-down reactions for mass spec were performed using SDS. I am not aware of published data on the effects of SDS on this class of proteins.

Thank you for pointing this out. In this new version of the manuscript, we now comment on the fact that detergents may artificially promote BCL-XL interactions, further justifying our use of resonance energy transfer techniques to document and investigate BCL-XL/KRAS interactions in the absence of cell lysis.